



**On the sensitivity of meteorological forcing resolution on hydrologic metrics**
Fadji Z. Maina[1*], Erica R. Siirila-Woodburn[1], Pouya Vahmani[2]
[1] Energy Geosciences Division, Lawrence Berkeley National Laboratory 1 Cyclotron Road, M.S.
74R-316C, Berkeley, CA 94704, USA
[2] Climate and Ecosystem Sciences Division, Lawrence Berkeley National Laboratory 1
Cyclotron Road, M.S. 74R-316C, Berkeley, CA 94704, USA
*Corresponding Author: fadjimaina@lbl.gov





**Abstract**
Projecting the spatio-temporal changes to water resources under a no-analog future climate
requires physically-based integrated hydrologic models, which simulate the transfer of water and
energy across the earth's surface. These models show promise in the context of unprecedented
climate extremes given their reliance on the underlying physics of the system as opposed to
empirical relationships. However, these techniques are plagued by several sources of uncertainty,
including the inaccuracy of input datasets such as meteorological forcing. These datasets, usually
derived from climate models or satellite-based products, typically have a resolution of several
kilometers, while hydrologic metrics of interest (e.g. discharge, groundwater levels) require a
resolution at much smaller scales. In this work, a high-resolution watershed model is forced with
various resolutions (0.5 to 40.5 km) of meteorological forcing generated by a dynamical
downscaling analysis based on a regional climate model (WRF) to assess how the uncertainties
associated with the spatial resolution of meteorological forcing affect the simulated hydrology.
The Cosumnes watershed, which spans the Sierra Nevada and Central Valley interface of
California (USA), exhibits semi-natural flow conditions due to its rare un-dammed river basin
and is used here as a testbed to illustrate potential impacts on snow accumulation and snowmelt,
surface runoff, infiltration, evapotranspiration, and groundwater levels. Results show that
localized biases in groundwater levels can be as large as 5-10 m and that other metric biases (e.g.
*ET* and snowpack dynamics) are seasonally and spatially-dependent, but can have serious
implications for model calibration and ultimately water management decisions.

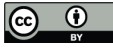

## 1. Introduction

Understanding water and energy fluxes across the Earth and the atmosphere is important

to assess the impacts of climate change on water resources. Integrated hydrologic models,
solving water-energy interactions and transfers, across the lower-atmosphere, the land surface,
and the subsurface, constitute a unique way to analyze water resources in both time and space
and to project into no-analog future where empirical models are no longer valid. With the
advancement of computing power, these models (e.g. MIKE-SHE (Abbott et al., 1986),
HydroGeoSphere (Panday and Huyakorn, 2004), and ParFlow-CLM (Maxwell and Miller,
2005)) are becoming widely used with high-fidelity and high-resolution. However, they are
plagued by several sources of uncertainty. Accuracy and precision, as well as uncertainty
reduction of hydrologic models, are extensively discussed in the literature. However, more
attention is given to the physical representation of the phenomena occurring in the hydrological
systems (Beven, 1993; Beven and Binley, 1992; Liu and Gupta, 2007), the reduction of
uncertainties related to the hydrodynamic parameters (Gilbert et al., 2016; Janetti et al., 2019;
Maina and Guadagnini, 2018; Srivastava et al., 2014), and the numerical resolution of the
mathematical equations governing the physics of the environment (Belfort et al., 2009;
Bergamaschi and Putti, n.d.; Fahs et al., 2009; Hassane Maina and Ackerer, 2017; Miller et al.,
1998; Tocci et al., 1997). Nevertheless, integrated hydrologic models, in essence, require
multiple sources of input data such as hydrodynamic parameters, initial and boundary conditions,
meteorological forcing data, etc.

Meteorological forcing is essential to inform integrated hydrologic models about the

atmospheric dynamics and therefore constitute one of the main drivers of the simulated



hydrologic processes. Like the hydrodynamic parameters or the initial and boundary conditions,
these data are impacted by several sources of uncertainty, including the fidelity of the physics of
the atmospheric model as well as, the representativity of the spatial resolution at which they
occur. Meteorological forcing data are often obtained from field measurements, satellite-based
data-assimilation products (Cosgrove et al., 2003), or climate models (Hurrell et al., 2013;
Skamarock et al., 2001). Because, the recent integrated hydrologic models require many
meteorological variables (i.e., precipitation, temperature, wind speed, solar radiation, air pressure
and relative humidity) to better simulate the interactions between the atmosphere and the
subsurface environment (i.e. the aquifers), climate models and satellite-based products are the
most used due to the scarcity of measurements. Moreover, in the context of climate change, only
climate models can provide a spatial distribution of future meteorological conditions. Also,
integrated hydrologic models require high resolution forcing to ensure fidelity and accuracy and
meteorological variables such as precipitation, one of the most important data and key control of
hydrological models, are very heterogeneous especially in mountainous areas (Olsson et al.,
2014; Prein et al., 2013).

Impact of the spatial resolution of meteorological forcing notably precipitation on runoff

and streamflow is widely documented in the literature with studies relying on *(i)* empirical
hydrologic models with precipitation data coming from measurements (Arnaud et al., 2002;
Berne et al., 2004; Lobligeois et al., 2014; Nicótina et al., 2008; Schilling, 1991; Shrestha et al.,
2006; Tobin et al., 2011), satellite-based products (Koren et al., 1999; Ochoa-Rodriguez et al.,
2015; Vergara et al., 2013) and climate models outputs (Dankers et al., 2007; Kleinn et al., 2005)
and *(ii)* physics-based hydrologic models with precipitation data coming from measurements
(Elsner et al., 2014; Fu et al., 2011), satellite-based products (Eum et al., 2014; Haddeland et al.,



2006) and climate models outputs (Mendoza et al., 2016; Rasmussen et al., 2011). Also,
Rasmussen et al., (2011) study the impact of meteorological forcing on snow dynamics.

Nevertheless, previous studies were mostly focused on runoff and streamflow analysis,

lacking a complete analysis of all the hydrodynamic processes occurring at the watershed scale.
Moreover, the resolutions of the meteorological data (~km) used remain relatively coarse
compared to the scale of resolution of the hydrological models (~m). Hence, the objective of this
study is to investigate the impact of the spatial resolution of the meteorological forcing from
~km to ~m on the hydrologic processes occurring at the watershed scale using a physics-based
integrated hydrologic model. In other words, we seek to understand how the uncertainties
associated with the coarse spatial resolution of meteorological forcing propagate into the high-
resolution integrated hydrologic models and affect the output of interest.

We use ParFlow-CLM (Kollet and Maxwell, 2006; Maxwell, 2013; Maxwell and Miller,

2005) forced with the Weather Research and Forecasting (WRF) model (Skamarock et al.,
2008a; Skamarock and Klemp, 2008). ParFlow-CLM simulates subsurface and surface flows as
well as their interactions by solving the mixed form of the Richards equation (Richards, 1931)
and the kinematic wave equation respectively as well as the processes driving the transfer of
water and energy from the ground surface to the atmosphere using a community land model (Dai
et al., 2003). Therefore, the model allows to analyze in both time and space, all the hydrological
components of interest such as the distribution of pressure head which encompasses the
information on the water level in the river and the groundwater levels, the groundwater and
surface water storages, the evapotranspiration, the infiltration, and the snow dynamics. ParFlow-
CLM is widely used by the scientific community as it physically solves several hydrologic
processes and can run in a high-performance computing framework. WRF, on the other hand,



solves the physics governing the atmospheric dynamics using a nested domain configuration to
provide meteorological forcing data at different spatial resolutions for ParFlow-CLM.

Our study focuses on Cosumnes, a unique watershed located in Northern California,

USA. This region, one of the largest in the United States, has unfortunately begun to experience
the effects of climate change. These effects are characterized by a fluctuation between extreme
drought causing unprecedented wildfires and periods of intense precipitation mainly caused by
atmospheric rivers. These rivers, a filament of concentrated moisture in the atmosphere, generate
storms with intensity much higher than the average and sometimes very localized. It is, therefore,
urgent to better understand how the water resources of this region evolve in response to these
uncommon conditions. Understanding water resources evolution is crucial to sustaining
California's agriculture ranked among the highest in the World. Assessing California's
hydrodynamics requires models that not only take into account the strong variations in
topography and land cover and land use, but also the snow dynamics. Indeed, the majority of the
water resources in this region originates from snowmelt. Also, because complex physics governs
the hydrology of the state such as sharp variation of wetting front accurate and high-resolution
models are necessary. As the region is characterized by both strong variations of weather and
complex hydrodynamics, it is, therefore, a good candidate to study how the spatial resolution of
meteorological forcing impact the hydrologic processes. The Cosumnes, a rare large-scale
watershed as it hosts one the last river without a dam in the state offering the opportunity to
study natural flow. Interestingly, the watershed is also capturing all the complex hydrologic
processes occurring in a typical Californian watershed including snow melting, groundwater, and
overland flow as well as, their interactions, bedrock hydrodynamics, strong spatial variation of
land use and land cover and topography. We study the water year 2017, the wettest water year on



California record characterized by several atmospheric rivers. The developed integrated hydrologic model has a spatial resolution of 200 m and we use five different spatial resolutions (40.5, 13.5, 4.5, 1.5 and 0.5 km) of meteorological forcing derived from the WRF dynamical downscaling approach. Our study aims to answer the following questions:

- What is the effect of the spatial resolution of meteorological forcing on the simulated snow accumulation and melt, evapotranspiration, infiltration and pressure head and/or water table depth? In broader terms, how meteorological uncertainties propagate into the resolved hydrodynamics and which processes require high-resolution meteorological forcing?

- At which spatial resolution should the climate models be solved to accurately describe the strong variations in meteorological conditions induced by atmospheric rivers and their effect on the hydrology and therefore water supply?

The section 2 of this manuscript describes the study area, section 3 is dedicated to the mathematical models used (ParFlow-CLM and WRF), and section 4 describes the results and the discussion of these findings.

## 2. The Cosumnes watershed model

### a. Study area

Located in Northern California, USA, the Cosumnes watershed is approximately 7,000 km$^2$ in size (Figure 1a) and hosts one of the last rivers in the region without a major dam. Thus, it offers a rare opportunity to study the natural flow conditions. The geologic composition consists of materials ranging from nearly impermeable formations (volcanic and plutonic rocks located mainly in the Sierra Nevada Mountains) to highly porous and permeable aquifers in the Central



Valley. The study area shows complex topographic patterns with elevations comprised between
the 2000 m and the sea level and strong variations of land use and land cover. The agricultural
region of Central Valley located in the southwest of the watershed consists of various crop types,
including alfalfa, pasture lands, and vineyards. These agricultural regions are subject to seasonal
pumping and irrigation while the Sierra Nevada Mountains are covered by predominately
evergreen forest. Spatial patterns of precipitation are highly heterogeneous across the watershed.
On average, the Sierra Nevada Mountains receive three times more precipitation (1500 mm) than
the Central Valley (Cosgrove et al., 2003), primarily in the form of snow. The regional climate is
considered Mediterranean, with wet and cold winters (with a watershed average temperature
equal to 0 °C) and hot and dry summers (with watershed average temperature reaching  30 °C)
(Cosgrove et al., 2003).





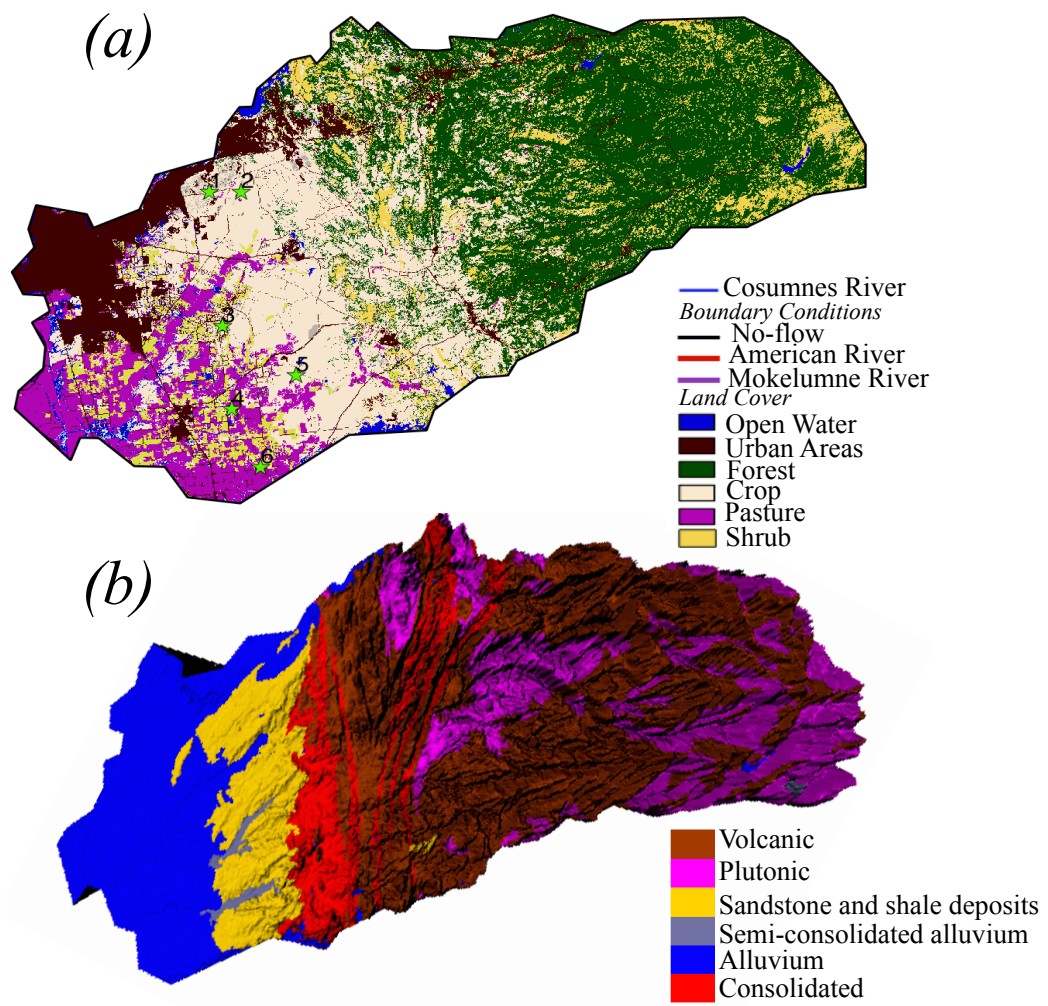

Figure 1: *(a)* Land-use and land-cover (Homer et al., 2015) and *(b)* geology (Jennings et al., 1977) and topography (USGS) of the Cosumnes Watershed

## 3. Numerical Modeling Methods

In this section, we briefly describe the two numerical models that we used in this study: (1) ParFlow-CLM, which simulates interactions as well as the transfer of water and energy between the lower atmosphere, the land surface, and the subsurface, and (2) Weather Research



Forecast (WRF), which simulates mesoscale numerical weather prediction, and is used here to
drive the meteorological conditions of the ParFlow-CLM simulations.
**3.1. Integrated Hydrologic Model: ParFlow-CLM**
ParFlow-CLM (Kollet and Maxwell, 2006; Maxwell, 2013; Maxwell and Miller, 2005)
describes the movement of water in the subsurface by solving the three-dimensional mixed form
of Richards equation (Richards, 1931), given by:
$$S_S S_W(\psi_P)\frac{\partial \psi_P}{\partial t} + \phi \frac{\partial S_W(\psi_P)}{\partial t} = \nabla.\left[k(x)k_r(\psi_P)\nabla(\psi_P - z)\right] + q_s \tag{1}$$

Where $S_S$ is the specific storage (L$^{-1}$), $S_W(\psi_P)$ is the degree of saturation (-) associated
with the subsurface pressure head $\psi_P$ (L), $t$ is the time, $\phi$ is the porosity (-), $k_r$ is the relative
permeability (-), z is the depth (L), $q_s$ is the source/sink term (T$^{-1}$), and $k(x)$ is the saturated
hydraulic conductivity (L T$^{-1}$). The interdependence of variables (i.e. relationships between
saturation and pressure head and between relative permeability and pressure head) is described
by the Van Genuchten model (van Genuchten, 1980). Overland flow is described by the two-
dimensional form of the kinematic wave equation given by:
$$-k(x)k_r(\psi_0)\nabla(\psi_0 - z) = \frac{\partial \|\psi_0, 0\|}{\partial t} - \nabla.\vec{v}\|\psi_0, 0\| - q_r(x) \tag{2}$$

Where $\|\psi_0, 0\|$ indicates the greater term between $\psi_0$ the surface pressure-head and 0, $\vec{v}$
is the depth averaged velocity vector of surface runoff (L T$^{-1}$), $q_r$ represents rainfall and
evaporative fluxes (L T$^{-1}$). The depth of the ponding water at the surface in $x$ direction ($v_x$) and $y$
direction ($v_y$) is calculated by:
$$v_x = \frac{\sqrt{S_{f,x}}}{n}\psi_0^{2/3} \quad \text{and} \quad v_y = \frac{\sqrt{S_{f,y}}}{n}\psi_0^{2/3} \tag{3}$$

Where $S_{f,x}$ and $S_{f,y}$ are the friction slopes in the $x$ and $y$ directions (respectively), and $n$ is
the manning coefficient.


192   Resolutions of the Richards and kinematic wave equations require the terms $q_s$ and $q_r(x)$

193 respectively. These terms include all the land surface processes simulated by CLM such as

194 evapotranspiration, infiltration, and snow dynamics. To compute these processes CLM uses the

195 soil moisture calculated by ParFlow, the vegetation characteristics (the type of land cover as well

196 as the physical properties of the plants) and the meteorological forcing calculated by WRF.

197   The Cosumnes ParFlow-CLM model is horizontally resolved at 200 m and varies in

198 vertical from 10 cm at the land surface to 30 m at the bottom of the domain. The total thickness

199 of the domain is 80 m. An analysis of variations in measured groundwater levels showed that this

200 thickness is sufficient to capture water table depth fluctuations and that in general, beyond 50 m

201 below the ground surface, the aquifer remains fully saturated. Simulations utilize parallel high-

202 performance computing to accommodate the large number of cells (approximately 1.4 million)

203 that constitute the high-resolution model.

204   The Cosumnes watershed is bounded by the American and Mokelumne rivers and is

205 constrained in the model with the use of weekly-varying values of Dirichlet boundary conditions

206 along these borders. A no-flow (i.e. Neumann) boundary condition is imposed at the eastern,

207 headwater side of the watershed. Hydrodynamic properties (including hydraulic conductivity,

208 specific storage, porosity, Van Genuchten parameters) are derived from a regional geological

209 map (Geologic Map of California, 2015; Jennings et al., 1977) and previous studies (Faunt et al.,

210 2010; Faunt and Geological Survey (U.S.), 2009; Flint et al., 2013; Gilbert and Maxwell, 2017;

211 Welch and Allen, 2014).

212   The 2011 National Land Cover (NLCD) map (Homer et al., 2015) is used in CLM to

213 define land use and land cover. Agricultural maps provided by the National Agricultural

214 Statistics Service (NASS) of the US Department of Agriculture's (USDA) Cropland Data Layer





(CDL) (Boryan et al., 2011) are further used to delineate specific croplands in the Central Valley.
Vegetation parameters are defined by the International Geosphere-Biosphere Programme (IGBP)
database (IGBP, 2018). Pumping and irrigation rates are estimated in the model because a
comprehensive dataset of such rates in the Central Valley does not exist. Water demand is
calculated based on the average parcel size, the crop type, and the country. Irrigated water is
internally sourced from either nearby groundwater pumping wells or, if adjacent to a river,
surface water diversions. This allows for the mass conservation of water within the model.
Fractions of water use between groundwater pumping and river diversions have been determined
by the California Department of Water Resources (DWR) (California Department of Water
Resources, 2010) and the United States Geological Survey (USGS) (USGS, 2018) databases, and
are used here. A seasonal pumping and irrigation cycle is assumed to be from April to November
based on the regional climate and discussions with local stakeholders.

A full water year is simulated to demonstrate how different scales of meteorological

forcing impact both wet and dry seasons of the year.  The water year 2017 (i.e. October 1st, 2016-
September 30th, 2017), a particularly wet year, is selected to conservatively demonstrate how
forcing scales may impact hydrologic results in a wide range of weather conditions. Initial
conditions of pressure head are derived from a longer simulation 2012-2017 performed with a
calibrated and spun-up model.

**3.2. Meteorlogical Model: WRF**

WRF (Skamarock et al., 2008b; Skamarock and Klemp, 2008) is a state-of-the-art, fully

compressible, non-hydrostatic, mesoscale numerical weather prediction model. The
parametrizations that represent physical processes in the configuration of WRF used here include





the Dudhia scheme (Dudhia, 1988) for shortwave radiation, the Rapid Radiative Transfer Model
(Mlawer et al., 1997) for longwave radiation, the Morrison double-moment scheme (Morrison et
al., 2009) for microphysics, University of Washington (TKE) Boundary Layer Scheme
(Bretherton and Park, 2009) for the planetary boundary layer, and the Eta Similarity scheme
(Monin and Obukhov, 1954) for the model surface layer. The Grell-Freitas scheme (Grell and
Freitas, 2014) is used for cumulus parameterization in two outer-most domains only (d01 and
d02). For domain d03 and d04, the higher-resolutions allow for convection to be resolved
explicitly. Mass balance validation results are shown in Appendix A. The described
configuration of WRF has been extensively validated against ground observation of
meteorological conditions in the California region in previous work (Vahmani et al., 2019;
Vahmani and Jones, 2017).

As shown in Figure 2, we configure WRF version 3.6.1 over four two-way nested

domains with a horizontal resolution of 13.5 km (domain 1), 4.5 km (domain 2), 1.5 km (domain
3), and 0.5 km (domain 4). Each domain is composed of 30 vertical atmospheric levels. Land
cover in WRF matches the one used in ParFlow-CLM. Post-spin-up soil moisture from ParFlow-
CLM is used to initialize the WRF model at the beginning of the simulation. Other WRF initial
conditions, as well as boundary conditions, are based on the NLDAS-2 forcing data set. Using
the nested domain configuration of WRF described above, we design a series of simulations to
dynamically downscale across the four spatial resolutions. The coarsest scale of forcing at 40.5
km resolution is generated by statistically up-scaling the coarsest of the WRF simulations (13.5
km). WRF simulations are conducted from September 1[st], 2016 to September 30[th], 2017,
covering the entire water year 2017 plus one month of spin-up. Spatial distributions of



precipitation and temperature at selected times obtained with the five spatial resolutions of
forcing are shown in Appendix A.

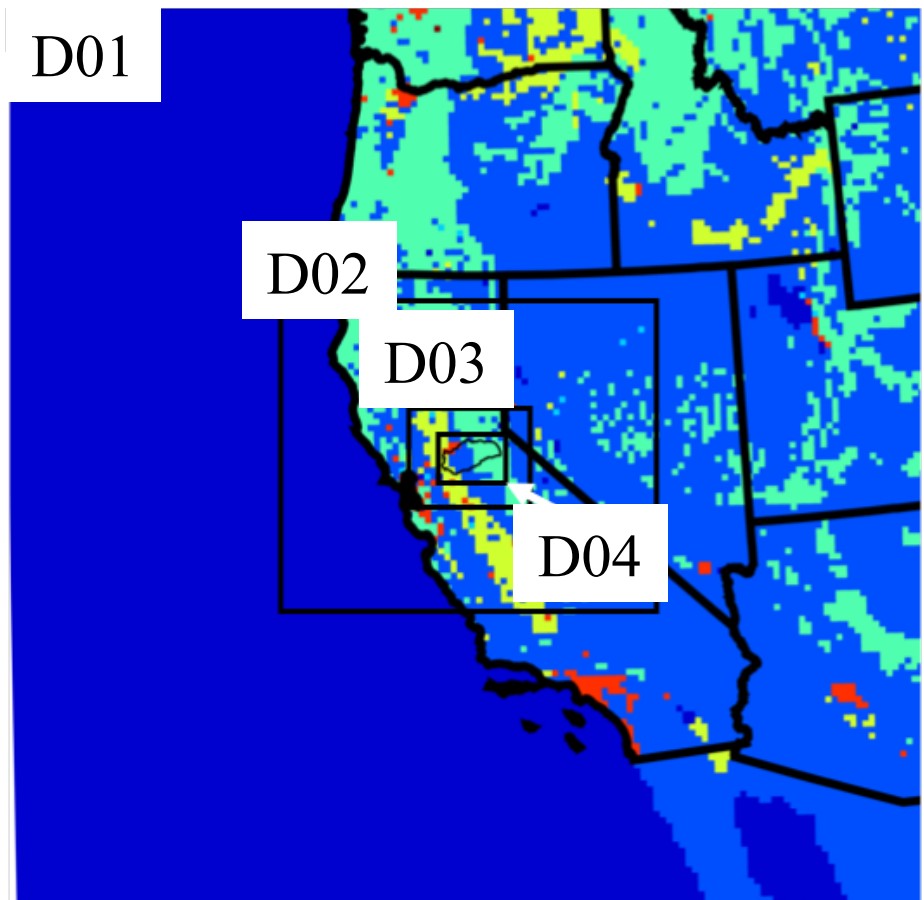


Figure 2: Land cover map (Homer et al., 2015) and geographical representation of four WRF
nested domains with 13.5, 4.5, 1.5, and 0.5 km spatial resolutions for d01, d02, d03, and d04,
respectively.

**3.3.Hydrologic metrics**

Results from the 5 spatial resolutions are compared for key land surface and subsurface


processes. We consider the results obtained with the finest spatial resolution of meteorological
forcing (0.5 km, closest to that of the hydrologic model) as the exact resolution, and evaluate the
differences relative to that of the 4 remaining resolutions (1.5, 4.5, 13.5 and 40.5 km).
Comparisons are shown as an absolute error (*AE*) and/or percent error (*PE*) relative to the 0.5 km
results via:
$$AE_{i,t} = X_{0.5_{i,t}} - X_{R_{i,t}} \qquad (5)$$
and
$$PE_{i,t} = \frac{X_{0.5_{i,t}} - X_{R_{i,t}}}{X_{0.5_{i,t}}} \times 100 \qquad (6)$$
where *X* is the model output (*ET*, Infiltration *I*, *SWE*, or pressure head, $\psi$) at a given point in
space (*i*) at a time (*t*), and *R* is the spatial resolution of the forcing (1.5, 4.5, 13.5 or 40.5 km).
Snap-shots in time of these errors highlight the sensitivity of each scale of forcing in space.
Global (i.e. domain-wide) differences are also calculated for select parameters of interest and
shown as a function of time by taking a domain average of each cell-based value of $AE_{i,t}$ or
$PE_{i,t}$ within the watershed.

Because large-scale changes in storage are of interest from a water management

perspective, total surface water (SW) storage is calculated via:
$$Storage_{SW} = \sum_{i=1}^{n_{SW}} \Delta x_i \times \Delta y_i \times \psi_i \qquad (7)$$

where $n_{SW}$ is the total number of river cells (-), $\Delta x_i$ and $\Delta y_i$ are cell discretizations along

the x and y directions (L), and *i* indicates the cell. Note that because ParFlow-CLM is an
integrated hydrologic model, only surface cells whose pressure head is greater than zero are
taken into account in the above summation. Similarly, total groundwater (GW) storage is
calculated via:



$$Storage_{GW} = \sum_{i=1}^{n_{GW}} \Delta x_i \times \Delta y_i \times \Delta z_i \times \psi_i \times \left(S_{s_i}/\phi_i\right) \qquad (8)$$
where $n_{GW}$ is the total number of subsurface saturated cells (-) and $\Delta z_i$ is the
discretization along the vertical direction the cell (L).

**4. Results and discussions**
**4.1.Snow Water Equivalent, *SWE***
In this watershed characterized by strong topographic variations and a large amount of
precipitation falling in the form of snow in the upper part of the watershed, it is crucial to
analyze how the different spatial resolutions of forcing data affect snow dynamics, a key control
of the hydrodynamics in the Central Valley aquifers. First, we compare the total *SWE* at the
watershed scale obtained with the 5 resolutions (see Figure 3). Our results indicate that all the
four resolutions overestimate the *SWE* when compared to the results obtained with 0.5 km
forcing and that there is a large difference in *SWE* spatial resolution depending on the scale of
the forcing used. We note that the accumulation of *SWE* starts at the same time for all resolutions
while the time of snowmelt varies considerably from one resolution to another. The coarser a
resolution is, the more the snowmelt timing is delayed. For example, *SWE* results obtained with
the 40.5 km resolution forcing exhibits low global error for the first half of the water year during
to snow accumulation, however during ablation the differences are very large both in terms of
magnitude (*PE* = 90 %) and timing (which is delayed by around 40 days). Due to the complexity
of the snow dynamics, in addition to the strong variations in the topography of our study area,
the results show that *SWE* is very sensitive to the spatial resolution of the meteorological data,
and that an accurate representation of *SWE* requires forcing data that is of similar resolution to
that of the hydrologic model. These conclusions are somewhat different from those drawn by



(Rasmussen et al., 2011), who found that the representation of *SWE* in mountainous systems can
be accurate for spatial resolutions of forcing lower than 6 km. A possible explanation for this
difference is the resolution of the physics-based model used in this study compared to that of
Rasmussen and co-authors and potentially to the varied complexity of the two simulated
watersheds and models.

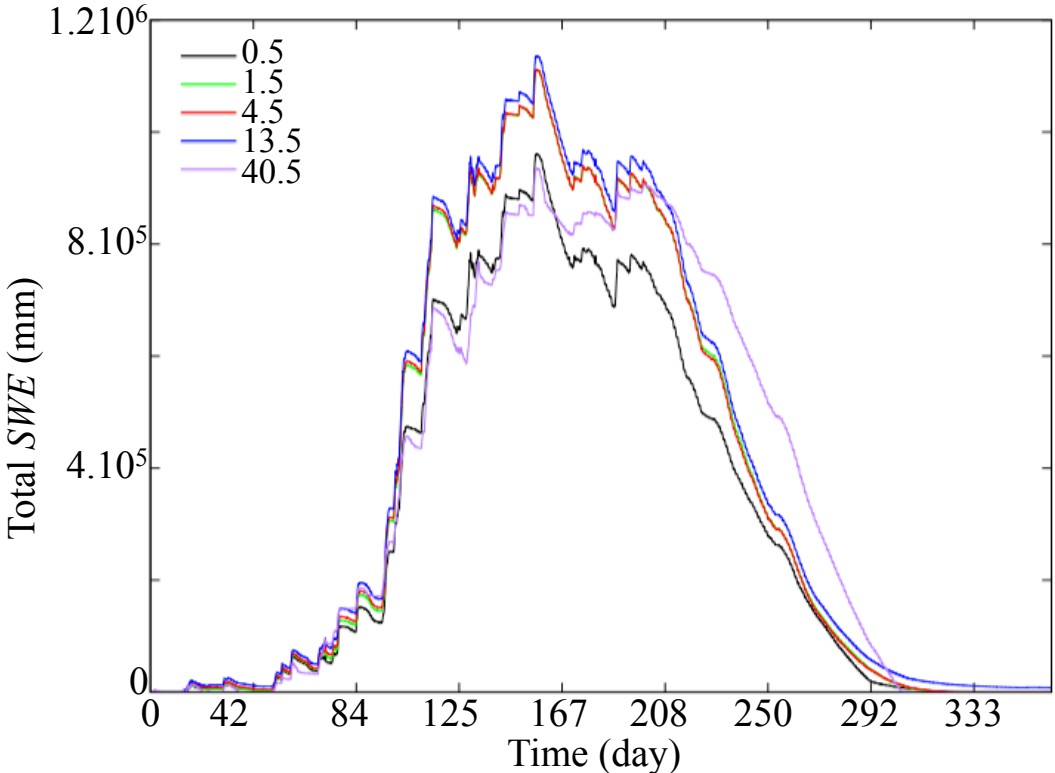


Figure 3: Temporal variations of the total Snow Water Equivalent *(SWE)* obtained with
meteorological forcing at spatial resolutions of 0.5, 1.5, 4.5, 13.5, and 40.5 km.

Figure 4a shows that the spatial distribution of *SWE* is more accurate for high-resolution

meteorological data and that the Cosumnes watershed forcing resolutions at and above 13.5 km


the extent of the watershed covered by snow is not well estimated. Figure 4b shows that while
the errors in *SWE* distribution certainly decrease with increasing the resolution of the forcing
data, errors remain relatively high (on the order of $AE = 100$ mm). However, the partition
between the areas of over and under- estimation appears to be uniform, for the *SWE,* we notice
that these zones also depend on the topography. This is because the snow processes depend not
only on the meteorological conditions but also on the slope and aspect. Depending on the
elevation, the orientation of the cell (north and south facing), the energy fluxes are different
resulting in very different snow dynamics. This strengthens the conclusions drawn previously
stating that the meteorological data should be at the resolution of the input data as well as the
physics-based model to ensure a good precision and accuracy in the representativity of the snow
dynamics.
*(a)*

*(b)*

Figure 4 Spatial distributions of *(a)* the *SWE* obtained with the five spatial resolutions of
meteorological forcing and *(b)* absolute error (*AE*) of *ET* with respect to the highest spatial
resolution of meteorological forcing (0.5 km). Results are shown at WY days 125 and 166.




### 4.2. Evapotranspiration, *ET*

*ET*, as shown here, is a combination of evaporation from the ground, canopy surfaces, transpiration by plants, and sublimation. Figure 5 shows the domain-average, temporal variation of the relative difference in the total *ET* flux as calculated with equation (5). Our results show that differences in spatial resolution on *ET* flux are mostly weak, and are only high after a storm event. The error generally increases as the resolution of the meteorological forcing increases. It is interesting to note, however, that for some time steps the relative differences obtained with the third coarsest meteorological forcing (13.5 km) are the largest. A possible explanation is the aggregated nature of the domain-average *ET*. Depending on the time step, the coarser forcing resolutions can lead to either an over or under- estimation of *ET*. Results do not show a systematic trend with regards to the over- or under- estimation of *ET*, where even at a single time-step, some resolutions indicate an overestimate of *ET*, others an underestimate. It is therefore difficult to establish a clear relationship between the spatial resolution and the directionality of *ET* error at a watershed scale. Note, however, that these errors do not increase over time. This can be related to the fast-changing nature of *ET* that is strongly linked to short-lived weather patterns and the diurnal cycle.

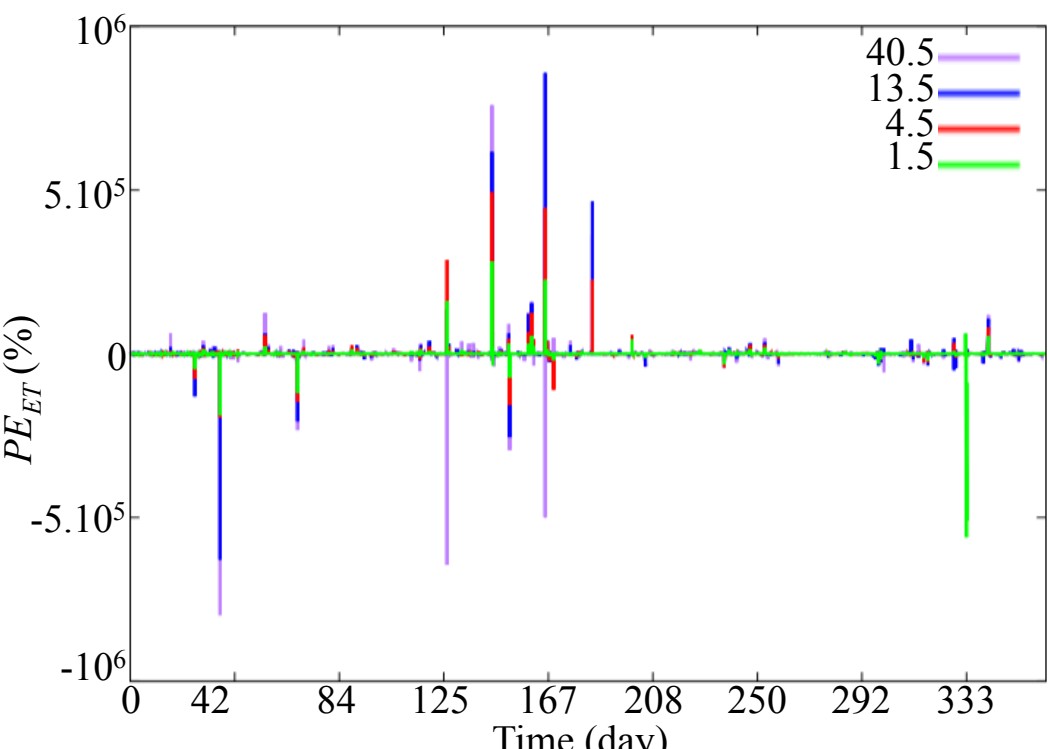


Figure 5: Temporal of the percent error (*PE*) of *ET* with respect to the highest spatial resolution
of meteorological forcing (0.5 km).

Figure 6a shows the spatial distribution of *ET* associated with the five resolutions at two

selected time steps (summer and winter). The spatial distribution of *ET* at these time steps is
different, and in general, all the five resolutions can distinguish these spatial differences of *ET* in
time. As expected, the most accurate *ET* distribution is obtained with the highest resolution of
the meteorological data, the coarser a resolution of meteorological data is the less accurate the
spatial distribution of *ET*. Because the results obtained with the high-resolution forcing is similar
to the resolution of the integrated hydrologic model (and thus the resolution of input data such as
topography, geology and land use and land cover), it allows us to better understand the



relationships between *ET* and these high-resolution data layers. Such analyses are difficult to
undertake for coarser resolutions.

*(a)*

| 0.5 km | 1.5 km | 4.5 km | 13.5 km | 40.5 km |

WY day 0

WY day 167

$$0.0 \quad 1.25 \ 10^{-5} \quad 2.5 \ 10^{-5} \quad 3.75 \ 10^{-5} \quad 5 \ 10^{-5}$$

$ET$ (mm/s)


*(b)*

| 1.5 km | 4.5 km | 13.5 km | 40.5 km |

WY day 0

WY day 167

$$-3000 \quad -1500 \quad 0 \quad 1500 \quad 3000$$

$PE_{ET}$ (%)


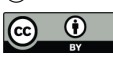



Figure 6: Spatial distributions of *(a)* the *ET* obtained with the five spatial resolutions of
meteorological forcing and *(b)* percent error (*PE*) of *ET* with respect to the highest spatial
resolution of meteorological forcing (0.5 km). Results are shown at the first day of the simulation
(WY day 0) and during the time at which peak differences are observed (WY day 167).

Seasonality and location impact the degree to which forcing scales impact *ET*. Note that

for the spatial distributions of *ET* associated with the second time step considered (day 167), the
results obtained with the five resolutions are very similar in the Central Valley. At this time
spatial patterns of *ET* only differ in the Sierra Nevada Mountains and the intrusion. The geology,
as well as, the land cover and the topography are more or less uniform in this valley, whereas
these parameters notably topography vary significantly in the Sierra Nevada Mountains. For the
first time step, the differences observed in the Central Valley are due to the fact that for very
precise resolutions of the forcing, the evolution of the storm is accurate (see Appendix A) and so
is the *ET*. Thus, for relatively homogeneous areas such as the Central Valley, high-resolution
forcing data is required only if the storm shows a strong spatial variation within the areas
whereas for highly heterogeneities associated with geology, topography, and land-cover, high-
resolution forcing data are always required if one is interested in analyzing accurately the spatial
distribution of *ET*.

Figure 6b shows the spatial distributions of percent error of *ET* relative to the results of

the 0.5 km meteorological forcing. Whatever the resolution considered, we note both an over-
and under- estimation of *ET* on the same scale of error (+/- 3000%), but with more localized and
less wide-scale differences at the finest scale of meteorological forcing. Also, as previously
noted, the error is higher in the Sierra Nevada Mountains than in the Central Valley for all





resolutions, especially later in the water year. This reinforces the conclusions drawn previously,
namely that for complex environments a precision in the meteorological data is strongly
required.

**4.3.Infiltration**

As shown in Figure 7a, the spatial resolution of forcing data strongly impacts the spatial

distribution of infiltration. Indeed, for coarse resolutions (i.e. 40.5 km), it is almost impossible to
determine the position of the storm and its impact on infiltration, the results obtained at this scale
are strongly dependent on the resolution of the forcing. However, for more precise resolution
(i.e. 0.5 km), we can exactly see the location of the storm, this resolution allows distinguishing
areas characterized by a very weak infiltration as the upper part of the catchment corresponding
to the Sierra Nevada Mountains. Indeed, in this area, due to the accumulation of snow
(precipitation is in the form of snow unlike in the Central Valley), the resulting infiltration is
zero. The spatial extension of the area subject to the snow accumulation is only accurate for
high-resolution meteorological forcing results.

*(a)*

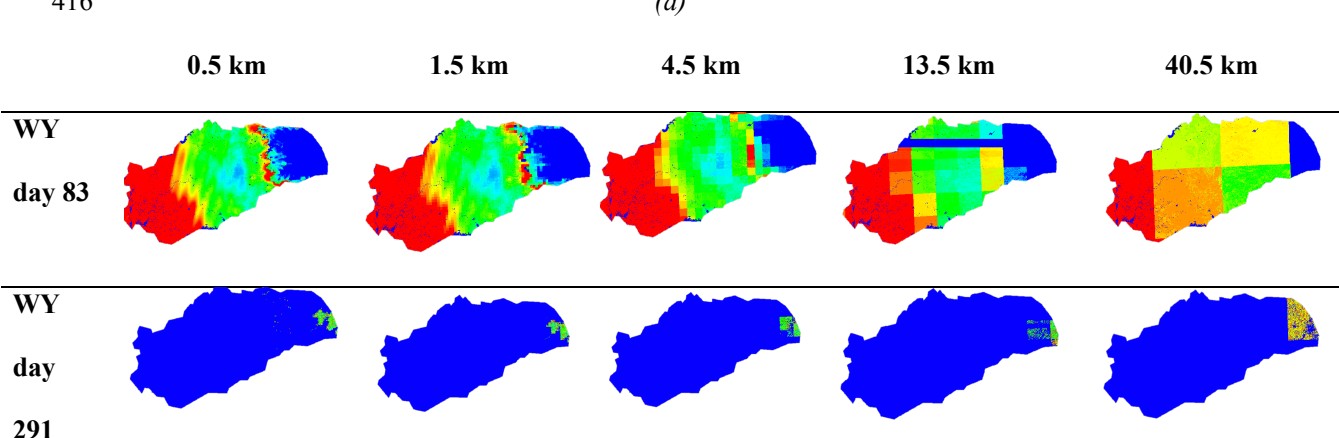



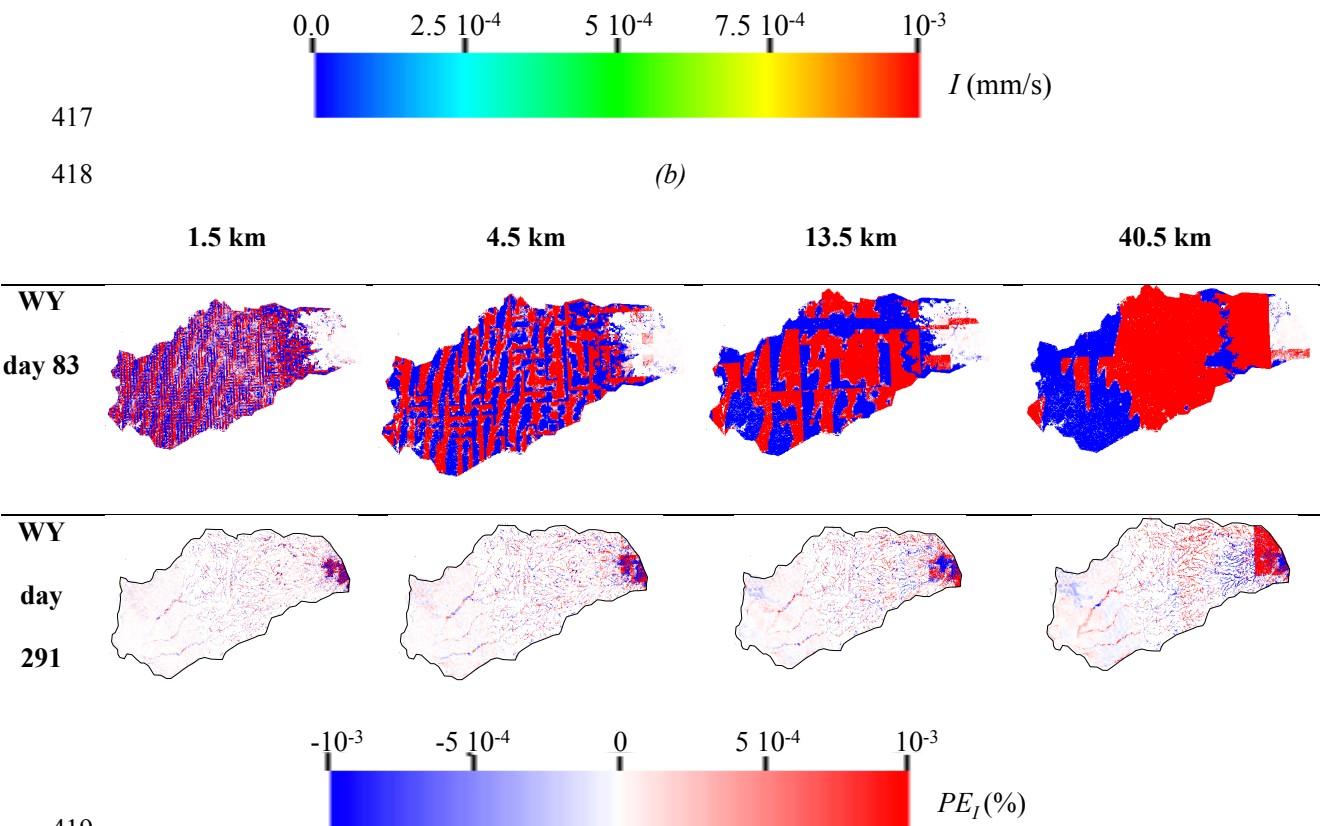


*(b)*

Figure 7: Spatial distributions of *(a)* infiltration *I* obtained with the five spatial resolutions of meteorological and the *(b)* percent error (*PE*) of infiltration *I* with respect to the highest spatial resolution of meteorological forcing (0.5 km). Results are shown in winter (WY day 83) and summer (WY day 291).

To better understand how the quality and precision of the spatial distribution of infiltration deteriorates by decreasing the resolution of the input data, we illustrate in Figure 7b, the spatial distribution of the percent error associated with the four resolutions considered at two selected time steps. These time steps show different dynamics. For the first time step corresponding to the period of snow accumulation, the errors are null in the Sierra Mountains

which is not the case for the second time step. Whatever the resolution considered, and as
previously discussed, we note that depending on the point considered there may be over- and
under- estimation of the infiltration and this is because the coarse resolutions represent an
average as explained previously. Note that these differences are observed over the entire
watershed except in the Sierra Mountains for the first time step, while for the second time step,
these errors are only observed along the river and its tributaries as well as in the Sierra Nevada
Mountains. This second time step corresponds to the summer, a snowmelt period and without
rain. As such, differences of infiltration in the Sierra Nevada Mountains are due to the snow
melting. As for the differences observed close to the areas subject to the overland flow, these are
due to the exchanges between the surface flow and the subsurface. Because the amount of snow
accumulated as well as the spatial extent of the area subject to snow dynamics is different for the
five resolutions considered, the resulting snowmelt is different. Thus, the runoff controlled by
this snowmelt will also be different and so is the infiltration of the quantities of water coming
from the overland flow. This indicates that the effects of the spatial resolution of forcing data can
be delayed in time.

**4.4. Surface and subsurface flow**

**4.4.1.   Surface water storage and river stage**

Figure 8 illustrates the *PE* between the highest resolution considered as the exact solution

and the other coarser resolutions. In general, the percent error is small (inferior to 5%) regardless
of the time of the year, and that these differences are almost zero for the results obtained with 1.5
and 4.5 km forcing resolutions for the entire water year. These errors are relatively small given
that some regions in the domain over-estimate pressure head and other regions under-estimate





pressure head (see Figure 9). In contrast, while the error is negligible at the beginning of the
simulation for results obtained with forcing at 13.5 and 40.5 km, the *PE* increases over time,
eventually reaching a nearly maximum at the end of the water year. This suggests that *PE* may
be cumulative and that longer simulations with overly coarse scales of forcing will compound
through time. It's interesting to also note that while the results obtained with the 13.5 km
resolution forcing overestimates the surface water storage at any time, the 40.5 km resolution
over-estimates at the beginning of the simulation and under-estimates at the end of the
simulation. Moreover, the errors obtained with the 13.5 and 40.5 km resolution are of the same
order but opposite signs. This suggests that although the total water budget is nearly equivalent
for each scale of forcing considered here (see Appendix A), an inaccurate spatial distribution of
forcing can lead to an inaccurate redistribution (and possibly a delay) of water and energy, and
hence different signals of surface water storage.



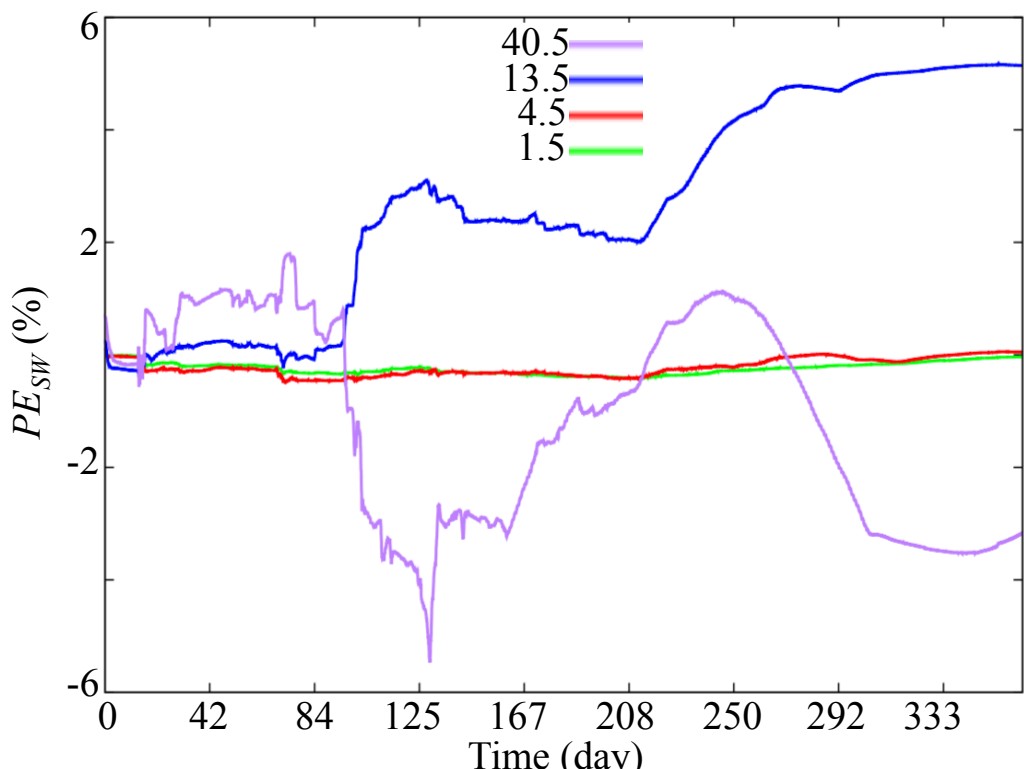

Figure 8: Temporal variations of the percent error (*PE*) of surface water storage relative to the
0.5 km forcing.

Figure 9 shows the spatial distributions of the absolute error associated with the pressure-
head of the first layer at two selected time steps. As mentioned in the preceding paragraphs, this
error increases with time, therefore, at the first time step the error is almost null for the spatial
resolutions of 1.5 and 4.5 km whereas it is non-zero for the second time step. Although the
spatial resolutions of 13.5 and 40.5 km have non-zero errors at the first time step, the error
increases considerably as the simulation proceeds. We note that the areas sensitive to the spatial
resolution of the meteorological forcing data are approximately the same for all four resolutions.





Indeed, the absolute error is null at the intrusion on contrary to the Central Valley and in the
Sierra Nevada Mountains. Interestingly, these two zones have different areas of influence, in the
Central Valley, the errors are non-zero everywhere except at the river, which is contrary to the
trend observed in the Sierras. This is related to the geological nature of these environments. Due
to the very low permeability and roughness of Sierra Nevada Mountains, any water from
precipitation will quickly contribute to surface runoff, which is highly sensitive to the spatial
resolution of forcing, on contrary to the Central Valley characterized by high permeability and
low manning coefficient and therefore low overland flow.

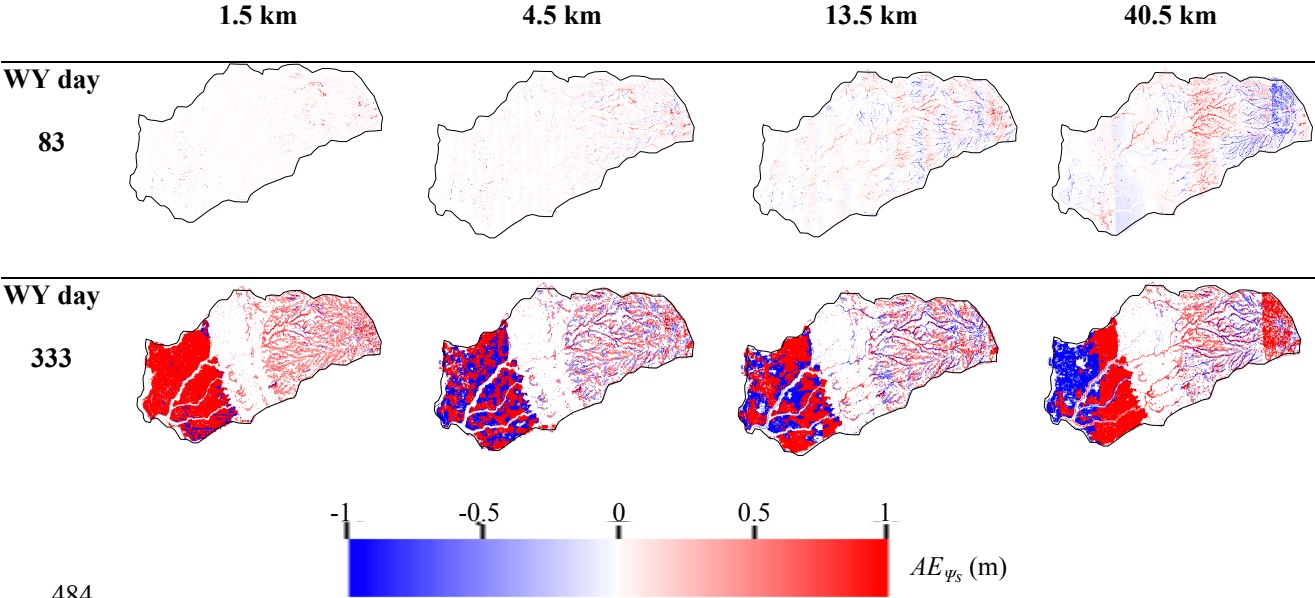


Figure 9: Absolute error (*AE*) of surface pressure-head ($\Psi_s$) with respect to the highest spatial
resolution of meteorological forcing (0.5 km). Results are shown in winter (WY day 83) and
summer (WY day 333).





Figure 10 shows the spatial distribution of the maximum difference in river water levels
between the results obtained with each spatial resolution of forcing and those obtained with the
0.5 km forcing. Maximum river water level differences are shown in absolute values (in units of
meters) and can occur at any point of time in the simulated water year. Differences in river water
levels at a given time step can reach 3 m. These differences are mainly located in the headwater
region of the watershed for results obtained with the finer resolution forcing progressively extend
into the Central Valley as the spatial resolution of forcing decreases. Our results suggest that
although the impact of spatial resolutions of forcing on the watershed-scale surface water storage
is low to insignificant (see Figure 8), at a given point in space and time differences may be
considerable. This can be especially problematic especially for calibration and validation
purposes because these methods adjust the input parameters of the model to reproduce the
measured water levels in the river with the model. In this case, differences between measured
and simulated values are not only due to parametric uncertainties but rather the forcing.



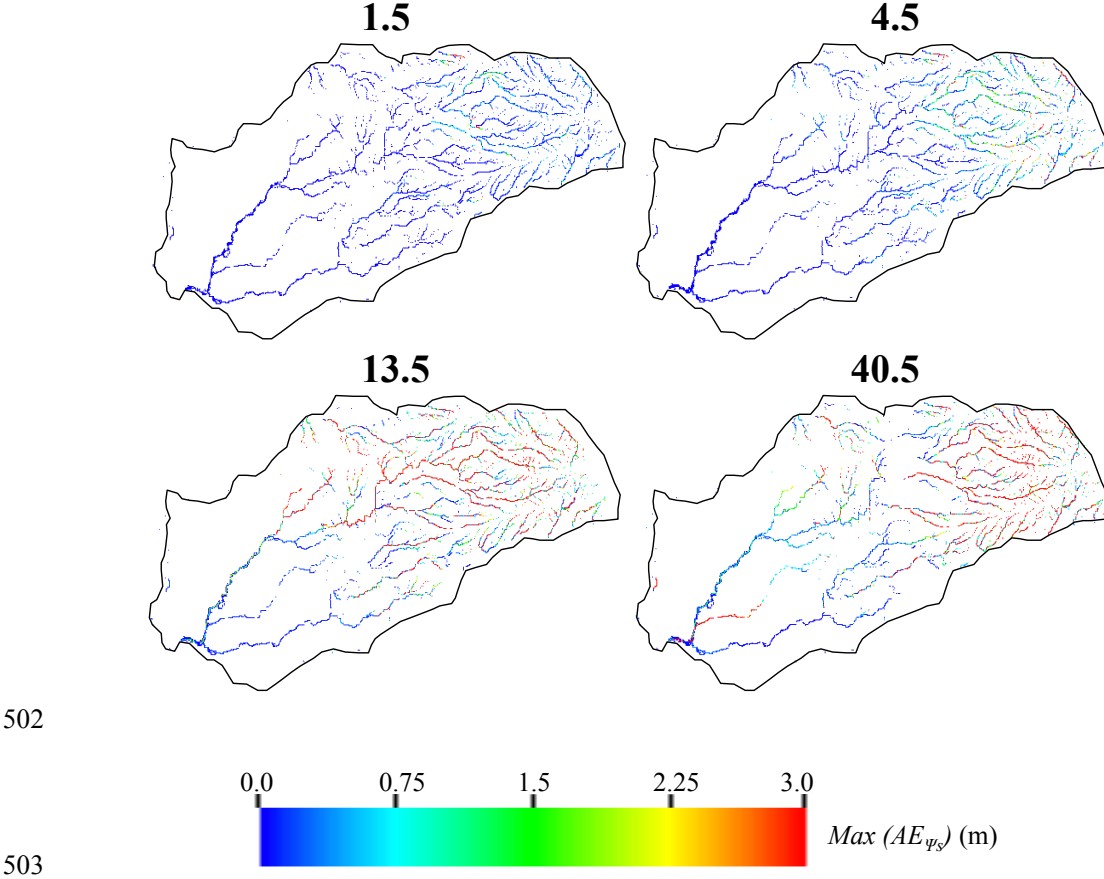


Figure 10: Spatial distributions of the maximum of Absolute Error ($AE$) in absolute
values of river height ($\Psi_s$) with respect to the highest spatial resolution of meteorological forcing
(0.5 km).

### 4.4.2. Groundwater storage and water table depth

For the cases considered here, the different spatial resolutions of forcing have very little
impact on the total groundwater storage of the watershed (Figure 11).

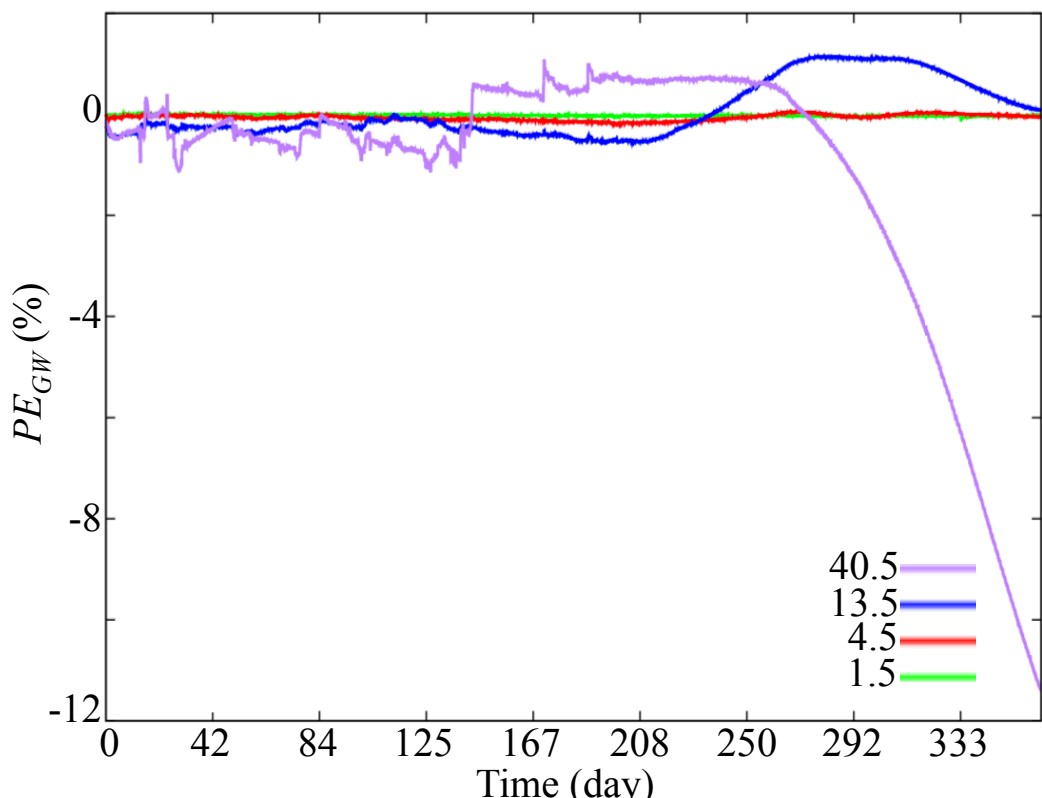

Figure 11: Temporal variations of the percent error (*PE*) of groundwater storage relative to the

0.5 km forcing.

Except the coarsest scale of forcing resolution towards the end of the simulation, the error

in groundwater storage for the different spatial resolutions of forcing yield very similar results.

The spatial resolution of 13.5 km overestimates the storage, however, this overestimation

remains very low of the order of 1% at certain times. In contrast, the groundwater storage results

obtained with the 40.5 km scale forcing are close to the exact solution at the beginning of the

simulation, yet reach error up to 10% at the end of the simulation. As stated previously, although

the total water budget associated with the meteorological forcing at the watershed scale is the





same for all the resolutions, the different spatial resolutions lead to different processes both in
time and space leading to different groundwater storages. Similar to the other maps of absolute
error, water table depth maps showing the *AE* relative to the results obtained with the 0.5 km
forcing show both over- and under- estimation of the water table depth as a function of the
forcing resolution (Figure 12a). As with the surface water storage, the groundwater storage error
is low due to the counterbalancing of opposite error signs. Note that we focused on the late time
step because for the first time steps these differences are too small to be used for interpretations.
For all the spatial resolutions considered, the Sierra Nevada Mountains are the most sensitive
areas to the spatial resolution of meteorological data, while the intrusion remains insensitive with
almost zero relative errors. This is due to the characteristics of the Sierra Nevada Mountains
which include strong variations of topography, snow dynamics, and impermeable rocks. The
intrusive zone is constituted of extremely impermeable materials so it has no groundwater
dynamics, as such the errors are zero. The spatial resolutions of 1.5 and 4.5 km have little impact
on the water table depth field associated with the Central Valley alluvial aquifers, the strong
relative errors are mostly observed for the results obtained with spatial resolutions of 13.5 and
40.5 km. Nevertheless, these errors are not uniform, they are marked along the river and outside
urban areas. As pointed out above, the hydrodynamics of the Central Valley depend on the Sierra
Nevada Mountains, whose snowmelt feeds the rivers and recharges the groundwater. The
absolute errors associated with the river areas are particularly due to the hydrodynamics of the
Sierra Nevada Mountains, in fact, as the snowmelt changes significantly according to the spatial
resolutions of the meteorological forcing considered as discussed in section 4.1, the exchanges
between the river and groundwater will thus be different. Note that these differences are also due
to the difference in evapotranspiration (section 4.2) and infiltration (section 4.3) and we highlight



that these differences accumulate over time as indicated by the errors that increase as the
simulation progresses.

*(a)*

|  |  |  |  |
|---|---|---|---|
| **1.5 km** | **4.5 km** | **13.5 km** | **40.5 km** |

-0.5      -0.25      0      0.25      0.5

$AE_{WTD}$ (m)



*(b)*

|  |  |  |  |
|---|---|---|---|
| **1.5 km** | **4.5 km** | **13.5 km** | **40.5 km** |

0.0      1.25      2.5      3.75      5

*Max (WTD)* (m)


Figure 12: Spatial distributions of *(a)* the absolute error (*AE*) of the water table depth (*WTD*)
with respect to the highest spatial resolution of meteorological forcing (0.5 km) at WY day 333,
and *(b)* the maximum of Absolute Error (*AE*) in absolute values of the water table depth (*WTD*)
with respect to the highest spatial resolution of meteorological forcing (0.5 km).




Figure 12b depicts the maximum differences (for all time steps) of the water table depth
in absolute value between the results obtained with the exact (highest) spatial resolution and the
other four spatial resolutions. As previously stated, due to the almost zero permeability of the
intrusion, the latter is insensitive to the spatial resolution of the meteorological data. The water
table depth differences are greater than 1 m in several places, particularly in the Sierra Nevada
Mountains. In the Central Valley, it should be noted that the strong differences are mainly
observed in the areas near the rivers and close to the pumping wells.
Figure 13 shows the temporal variations of the difference of the water table depth
between the highest resolution and the four other resolutions at 6 selected points. We selected
points located in the Central Valley as this zone hosts an alluvium aquifer (see their location in
Figure 1). For all these points, we note that the differences are almost zero for the spatial
resolution of 1.5 km indicating that this spatial resolution is sufficient to represent the
groundwater dynamics of this region. The spatial resolution of 4.5 km also shows relatively low
differences, the latter is indeed zero at three points and only the points 2, 4 and 5 have non-zero
differences, but these remain less than 50 cm. The strongest differences are observed for results
obtained with forcing spatial resolutions of 13.5 and 40.5 km; note that the coarsest resolution
does not necessarily give the highest differences. In fact, at points 4 and 5, the highest
differences are obtained with the resolution of 13.5 km, indicative of the complex over- and
under- estimation patterns of bias observed at these coarser resolutions of forcing. In general, the
use of these large-scale spatial resolutions of forcing can lead to an over- or under -estimation of
the pressure-head between 50 cm and 10 m.

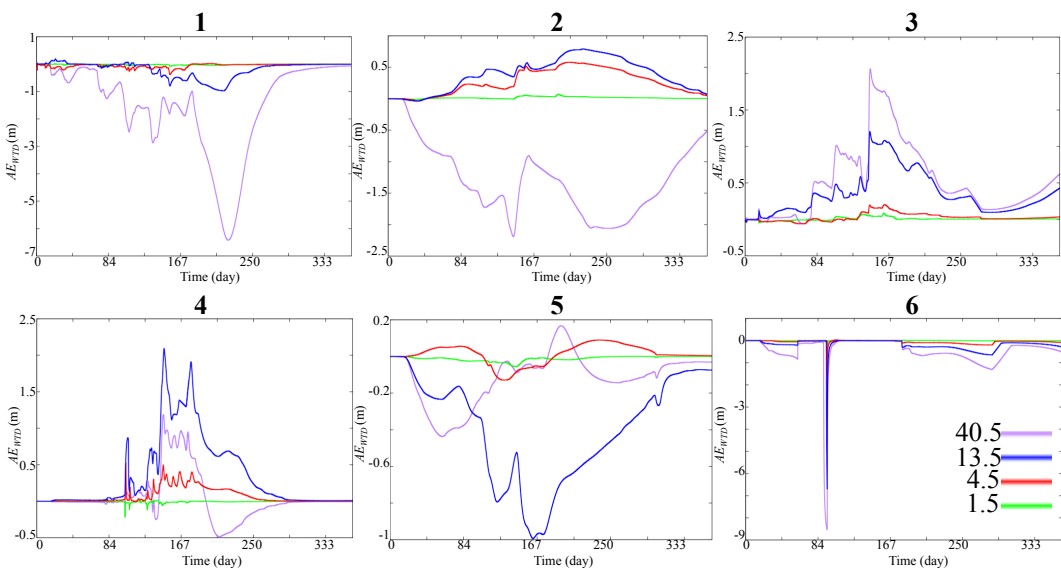

Figure 13: Absolute Error (*AE*) of the water table depth (*WTD*) with respect to the highest spatial resolution of meteorological forcing (0.5 km) at six selected points.

Thus, while our results indicate that the spatial resolution of meteorological forcing has little impact on the total groundwater storage, at discrete points within the watershed the spatial resolution of forcing is very important, especially for resolutions greater than 4.5 km in this watershed. Again, this is particularly an issue for model calibration purposes given that hydrologic numerical models are typically validated/calibrated by comparing the groundwater measurements with the model outputs. In this case, our results indicate that careful attention must be given to the spatial resolutions of forcing, as some errors are only due to the latter not to any model parameterization.





**5.  Conclusions**

Numerical methods that solve integrated hydrologic models are becoming increasingly

precise and of high-resolution. They thus require high-resolution and accurate input data such as
meteorological forcing. However, while integrated hydrologic models increase in precision, the
meteorological data used are most often of coarse resolution whereas these data are strongly
heterogeneous in space. It is, therefore, important to better understand not only how the
uncertainties associated with the spatial distribution of meteorological data affect the outputs of
hydrologic models, but also the spatial resolution of the meteorological forcing required to
minimize these uncertainties. Moreover, thanks to the development of atmospheric models, it is
now possible to obtain meteorological data at the same resolutions as the hydrologic models.

In this study, we used in a high-performance computing framework, the integrated

hydrological model ParFlow-CLM, to simulate the hydrodynamics of a complex and unique
watershed located in Northern California, the Cosumnes Watershed. Five different spatial
resolutions of meteorological data were obtained via the dynamical downscaling approach of the
Weather Research Forecasting (WRF) model. The Cosumnes watershed is an excellent candidate
to better understand how the different spatial resolutions affect the results of an integrated
hydrologic model of a watershed characterized by strong variations of topography, geology, land
use and land cover leading to highly heterogeneous and complex atmospheric and hydrologic
dynamics. The watershed allows also to investigate how the uncertainties related to the spatial
resolution of meteorological data affect the following key components of the hydrologic cycle:
snow dynamics, evapotranspiration, infiltration, surface and groundwater interactions, etc.

Our results show that the impact of the spatial resolution of meteorological data depends

on the hydrologic component of interest as well as the temporal and spatial scale.



- At the scale of the watershed, the total fluxes of evapotranspiration are more or less insensitive to the spatial resolution of forcing. However, to obtain an accurate distribution of evapotranspiration based on the physical properties of the watershed, a high-resolution forcing is required. Indeed, our results show that it is almost impossible to identify the change in evapotranspiration as a function of land use or geology with low-resolution meteorological data.

- The results obtained with infiltration are quite similar to those of evapotranspiration. Note that for these two processes, the relative errors induce by a coarser resolution obtained are most often marked after a storm, and that these errors automatically become very low as soon as the storm ends.

- In this watershed characterized by strong variations of topography, the errors associated with the spatial resolution of the meteorological data have a considerable impact on snow accumulation and melting, even at the scale of the watershed. The different spatial distributions obtained suggest that meteorological data with the same resolution as the hydrologic model is needed to accurately reproduce the distribution as well as the total volume of snow water equivalent. Unlike evapotranspiration and infiltration, where there is always an over- and under- estimation, for snow water equivalent, the relative errors obtained depend on both the spatial resolution and topography.

- The spatial resolution of the forcing data does not impact the total storage of the surface water at the watershed scale. Indeed, our results have shown that even for the coarsest resolution (i.e. 40.5 km), the error, increasing with time, is around 5%. However, we have emphasized that for the river levels at one point and at a



given time, the differences between the highest spatial resolution of the forcing
data and the four other resolutions can exceed 3 m. Our physical model has also
allowed us to determine areas such as the Sierra Nevada Mountains where runoff
is very sensitive to the spatial resolution of the weather data.
• We also obtained similar total groundwater storages at the watershed scale with
the five different spatial resolutions of the meteorological data. However at the
local scale, the variations of pressure head in the subsurface obtained with the
different resolutions are not the same, the differences can reach 9 m at a given
time and location, especially in the Central Valley alluvium aquifers.
Although the total water balance of the five spatial-resolutions of the meteorological
forcing is the same, the different spatial resolutions lead to different hydrological processes that
change both in time and space. For a good representation of the land surface processes
(infiltration, evapotranspiration and snow dynamics), a spatial resolution of the meteorological
data which is close to that of the hydrologic model is required due to the instantaneity and
complexities of these phenomena. For the surface and subsurface processes, we have
demonstrated that for this particular watershed, a spatial resolution of 4.5 km is sufficient to
reproduce precisely these mechanisms. As a result, satellite-based products such as NLDAS
resolutions may induce errors that may limit the use of these products for spatially accurate
studies. Because the coarse spatial resolutions may lead to very different groundwater and
streamflow variations compared to the highest resolution, particular attention must be paid to the
spatial resolution of meteorological data, especially in the calibration and/or validation processes
of numerical models. Indeed, the differences between the measured and simulated outputs are



not only due to the hydrodynamic parameters of the model but may also be related to the
parameterization of the meteorological data.

In this study, we have focused on the spatial distribution of meteorological data, future

studies will focus on the propagation of uncertainties related to the temporal resolution, and thus
determine the main source of uncertainties. Climate Models are also used for future climate
projections purposes, it would also be important to determine the ideal spatial-resolution of
forcing in this context.


**Code and Data availability**
Simulations inputs, models and data are available from the authors upon request.







## Appendix A

### *A.1 Mass Balance Validation*

The physics represented for the four WRF domains are identical, except for cumulus parameterization which is used for domains d01 (resolution of 13.5 km) and d02 (resolution of 13.5 km) and not for domains d03 (resolution of 1.5 km) and d04 (resolution of 0.5 km). The reason behind this is that WRF at resolutions higher than around 4 km (Gilliland and Rowe, 2007) can resolve convection explicitly. To assess the sensitiviy of the WRF simulated forcings to this inevitable incosistancy between the domains, we compare watershed-wide daily precipitation and air tempeature in figure XX. Our results show that there are minimal differences (RMSE of less than 0.002 m and 0.01 C for precipitation and temperature, respcetivly) between 4 WRF domains, when averaged over the watershed. This shows that the only difference between the forcings from WRF domains are due to different resolutions and the effects of described difference in physics representations are limited.



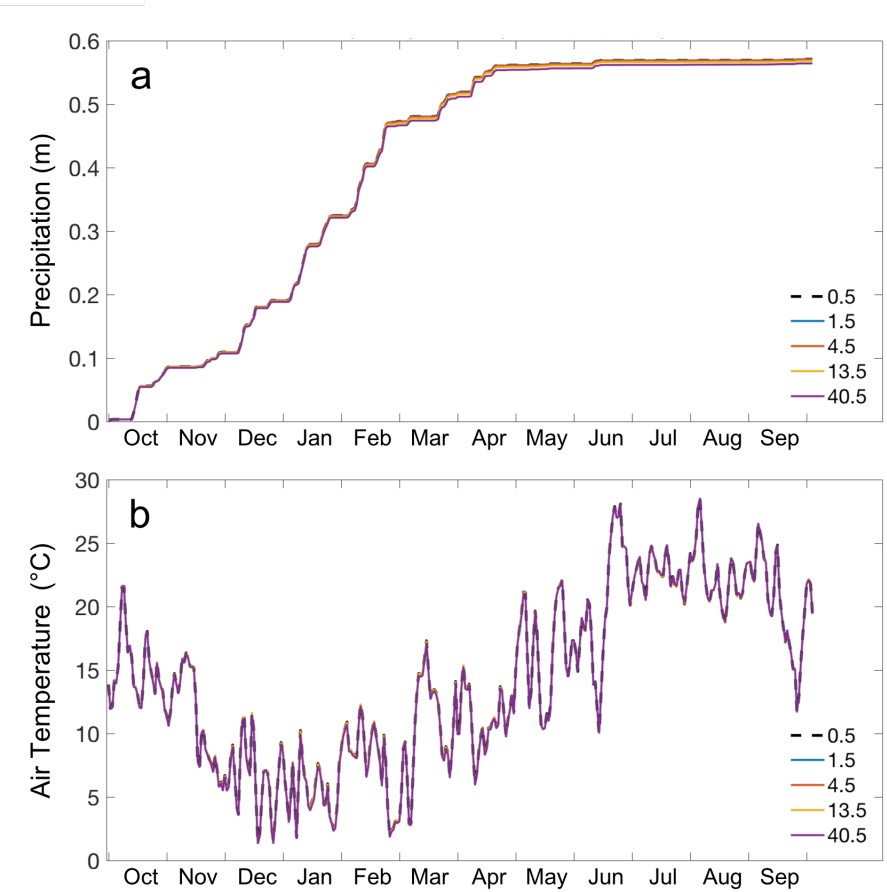


Figure A1: Daily variations of WRF simulated precipitation (a) and air temperature (b), averaged
over the entire watershed for spatial resolutions of 0.5, 1.5, 4.5, 13.5, and 40.5 km.





*A.2 Spatial distributions of precipitation and temperature over the domain d04*

| | WY day 1 | WY day 83 | WY day 125 |
|---|---|---|---|
| **40.5 km** | | | |
| **13.5 km** | | | |
| **4.5 km** | | | |
| **1.5 km** | | | |
| **0.5 km** | | | |

0,0      5.10⁻⁴      10⁻³    1.5 10⁻³    2 10⁻³

Precipitation (mm/s)



Figure A2: Spatial distributions of precipitation associated with the five spatial resolutions of
meteorological at three selected times






| | **WY day 1** | **WY day 83** | **WY day 125** |
|---|---|---|---|
| **40.5 km** | | | |
| **13.5 km** | | | |
| **4.5 km** | | | |
| **1.5 km** | | | |
| **0.5 km** | | | |

275.0    281.2    287.5    293.8    300

Temperature (K)





Figure A3: Spatial distributions of temperature associated with the five spatial resolutions of
meteorological at three selected times.


**Author contribution**
The authors contribute equally to this work.

**Competing interests**
The authors declare that they have no conflict of interest.

**Acknowledgements**
This work was supported by http://dx.doi.org/10.13039/ 100007000 (LDRD) funding from
Berkeley Lab, provided by the Director, Office of Science, of the U.S. Department of Energy
under Contract No. DE-AC02-05CH11231. This research used computing resources from the
National Energy Research Scientific Computing Center, a DOE Office of Science User Facility
supported by the http:// dx.doi.org/10.13039/100006132 of the U.S. Department of Energy under
Contract No. DE-AC02-05CH11231.
The authors are thankful to Peter-James Dennedy-Frank for his careful reading and constructive
suggestions and comments.



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
