# Peer review of "Sensitivity of meteorological forcing resolution on hydrologic variables Fadji Z. Maina1\*, Erica R. Siirila-Woodburn1, Pouya Vahmani2 1 Energy Geosciences Division, Lawrence Berkeley National Laboratory 1 Cyclotron Road, M.S. 74R-316C, Berkeley, CA 94704,"

_Hydrology and Earth System Sciences, 2019_

## Referee Comment (RC1) · Anonymous Referee #1 · 18 Dec 2019

General comments

This paper tested the sensitivity of the spatial resolution of meteorological forcing data on hydrologic model results. The paper addresses a classic scientific question which is within the scope of HESS. The descriptions of experiments and calculations are complete. However, the idea and findings of this paper are not novel. Moreover, the conclusions are derived from one meteorology model and one hydrologic model, the results to me are insufficient to support the conclusions. Lastly, the paper is not well written and the presentation of results analysis is unclear.

Specific comments

1. Supportability of the findings. As mentioned above, the conclusions are derived

from one meteorology model and one hydrologic model. These are not sufficient to provide a general conclusion. For example, the conclusion that "... the meteorological data should be at the resolution of the input data as well as the physics-based model to ensure a good precision and accuracy in the representativity of the snow dynamics." (Lines 333-335) may not be applicable if modelers adopt hydrologic response units, not grid cells, to build hydrologic models. For irregular computational unit based hydrologic modeling, what is the appropriate input data resolution? I hope to see more experiments or discussions on it.

2. Moreover, the accuracy of the WRF meteorology model and the downscale and upscale techniques are very important to this paper's results. The authors validated the quality of different resolution forcing data at the watershed lumped level and the distributed level in Appendix A. However, in Figure A.2 and Figure A.3, the meteorology at 40.5km look bad as the pattern becomes very blurry. I would encourage trying a finer resolution, such as 27km. In addition, is it possible to validate the WRF and downscaled meteorology with measured precipitation and temperature data for this research area? This will better valid your modelled and calculated meteorology forcing.

3. Title issue. It would be better to change "hydrologic metric" to something more appropriate. I suggest this because some hydrologists may understand "hydrologic metric" as hydrology model performance metrics, such as Nash–Sutcliffe efficiency (NSE) or Kling–Gupta efficiency. An alternative could be hydrologic prediction. Moreover, is the first word "on" redundant? Please correct me if I'm wrong.

4. Presentation of the results. The writing of this paper needs to be improved. I found it hard to understand some sentences, the definitions of several terms and the full name of an abbreviation are missing, and some citations and captions are not standardized. Please see below for details.

5. In section 2a, I suggest specifying the areas of the Sierra Nevada Mountains and the Central Valley in Figure 1.

[Figure]

Technical corrections

1. Line 268, "5 spatial resolutions". Change 5 to five. Similarly, to line 271.

2. Equation 7 and Equation 8, are the two $\varphi$ the same? I guess one is surface pressure head, and nother is subsurface pressure head. Please be consistent with the terms in Equation 1 and Equation 2, and be specific.

3. Line 341, citation (Rasmussen et al., 2011) should be Rasmussen et al. (2011).

4. Unclear sentences list. a. Lines 330-332 b. Lines 369-370 c. Lines 493-495 d. Lines 523-524 e. Lines 527-528 f. Lines 540-544 g. Lines 624-626 h. Lines 630-632

5. Incorrect caption citation list. a. Line 347, Equation 5 -> Equation 6. b. Line 686, figure xx. c. Line 687, 0.01 C.

6. Undefined abbreviation: Line 342, WY.

7. Unclear referent. a. Line 561, the latter. b. Line 590, the latter.

8. Line 666, Climate Models -> Climate models.

---

## Referee Comment (RC2) · Anonymous Referee #2 · 18 Dec 2019

General comments: The manuscript addresses the sensitivity of hydrologic variables to the spatial resolution of meteorological forcing inputs. Analysis of multiple components of the hydrological cycle in time and space makes the current study fit within the scope of HESS.  c While the overall objective is to compare how the resolution of meteorological forcing data impacts hydrologic variables, it would be helpful to see how WRF model output from the simulations compares to actual observations. The study year is said to be the wettest on record, so the WRF simulations are not being performed for a typical year, but rather one that lies in the tails of the distribution. As such, are there inherent errors associated with simulating an anomalous case versus typical? Comparison to observations may support whether WRF simulations are similar to reality, such that any biases are recognized before simply comparing model simulations

to one another. • Some of the background is given without any supporting sources. One example is the paragraph from lines 107-130. While some of the information may seem like common knowledge, it is still important to state where this information came from. For example, in lines 114-115, the authors state something is ranked "among the highest in the world." This is highest according to what? Similarly, lines 117-118 state that the majority of water resources in the region originate from snowmelt. How do we know this? Many readers will likely be familiar with such concepts, but those of inter-disciplinary backgrounds may not be. • Paper is generally well organized, however within the results section tends to jump around a bit. • Many of the paragraphs in the results section start with "Figure x shows..." This reads more like a listing of results based upon figures and seems to contribute to some of the disorganization within discussion of individual results. The results may be much clearer if figures were used to support the main points that the authors were trying to communicate rather than the authors trying to find main points to support each figure. Use the figures to tell your story and convince the reader of your main conclusions. It is difficult to keep track of whether figure A and figure B (not actual figures, just an example) both supported the same conclusion as many figures are discussed independently of one another. • The authors present two main questions to be addressed by the manuscript, however there was very little mention of how the results addressed question 2 (lines 137-139) within the conclusions. It isn't clear how everything ties together. The last statement in the conclusion paragraph is worded such that it seems to pose this question for future work rather than show how the current work addressed it.

Specific Comments: Lines 64-68: Reword if possible. Starting with "because" throws the reader off. Otherwise, if "Because" remains at the start the comma should be omitted, or use of the word "Because" should be omitted itself.

Lines 70-73: This could be split into two sentences to improve readability, otherwise I found myself trying to circle back to the beginning to understand what the point was. Splitting the two after "accuracy" seems logical.

Lines 82 and 83: Why does this study matter to the current work? It ties in later, I think, but it would be helpful to quickly tie your literature review into the current work.

Lines 114-115: Please reword. I'm not sure what exactly is ranked highest in the world. Is it the agricultural sustainability or the necessity for understanding water resources? Please also provide a source to support this conclusion.

Lines 127-128: What is the period of record when stating "wettest on record." How do the authors know this?

Lines 227-232: The authors state that 2017 is representative of a wide range of weather conditions, but also that it is a climatological anomaly. Why isn't a year that is more near-normal used to demonstrate an annual range of meteorological conditions? Swain et al. (2018) suggests a greater number of rapid transitions from dry to wet periods is likely in future climate scenarios. This may support the use of water year 2017 if the authors intentions were to capture a year that rapidly transitions from dry to wet but that is not clear in the current draft. This is the paper cited above: https://doi.org/10.1038/s41558-018-0140-y

Line 349: The authors state that error increases as resolution of the meteorological forcing increases. Does this mean that finer resolution, i.e., lower grid spacing results in greater error, or higher grid spacing (coarser resolution) does? Readers that are unfamiliar with numerical weather prediction models may get confused, and minor elaboration would ensure that the main point the authors are trying to communicate is not misinterpreted. It looks like this is touched on in lines 367 to 369, but it would be easier to interpret if these statements were all grouped together.

Lines 617-648: This part of the manuscript is well organized, and I appreciate how the main points are separated into individual bullets points.

Lines 630-632: The statement that the meteorological model should have the same resolution as the hydrologic model is made here, however, it does not appear that any

simulations of WRF were performed with the exact same resolution as the hydrologic model. Line 197 says that the horizontal resolution of the hydrologic model is 200 meters while the finest horizontal resolution WRF simulation was 500 meters, so how is this conclusion supported by the current work when there were still differences in resolution? If there are past studies that support such a conclusion, please add relevant citations.

Lines 666-668: This statement is somewhat unclear. Is this referring to what climate models do in general, or is this referring to how they will be employed in future work? Please reword the statement for clarity.

---

## Author Comment (AC1) · 26 Jan 2020

We thank the referee for acknowledging the scientific question highlighted in our work, as well as the experiments and the computations we performed. To the best of our knowledge, there aren't any existing studies, which utilize physics-based models to simulate both atmospheric and hydrologic processes in this coupled fashion with the goal to understand biases in resolution on hydrologic flow metrics. We have thoroughly revised the manuscript to highlight this novel approach and the importance of the study's findings. We have thoroughly revised the manuscript for clarity and worked to make the writing more succinct and clearer. We feel that these changes had improved the manuscript and made it more suitable for publication in HESS.

[Figure]

Please also note the supplement to this comment:
https://www.hydrol-earth-syst-sci-discuss.net/hess-2019-509/hess-2019-509-AC1-supplement.pdf

————————————————————

[Figure]

**Supplement:**

**Comments in plain text, response to comments in *blue italics*.**

**Anonymous Referee #1**

General comments

This paper tested the sensitivity of the spatial resolution of meteorological forcing data on hydrologic model results. The paper addresses a classic scientific question which is within the scope of HESS. The descriptions of experiments and calculations are complete. However, the idea and findings of this paper are not novel. Moreover, the conclusions are derived from one meteorology model and one hydrologic model, the results to me are insufficient to support the conclusions. Lastly, the paper is not well written and the presentation of results analysis is unclear.

*We thank the referee for acknowledging the scientific question highlighted in our work, as well as the experiments and the computations we performed. To the best of our knowledge, there aren't any existing studies, which utilize physics-based models to simulate both atmospheric and hydrologic processes in this coupled fashion with the goal to understand biases in resolution on hydrologic flow metrics. We have thoroughly revised the manuscript to highlight this novel approach and the importance of the study's findings. We have thoroughly revised the manuscript for clarity and worked to make the writing more succinct and clearer. We feel that these changes had improved the manuscript and made it more suitable for publication in HESS.*

Specific comments

1. Supportability of the findings. As mentioned above, the conclusions are derived from one meteorology model and one hydrologic model. These are not sufficient to provide a general conclusion. For example, the conclusion that ". . . the meteorological data should be at the resolution of the input data as well as the physics-based model to ensure a good precision and accuracy in the representativity of the snow dynamics." (Lines 333-335) may not be applicable if modelers adopt hydrologic response units, not grid cells, to build hydrologic models. For irregular computational unit based hy- drologic modeling, what is the appropriate input data resolution? I hope to see more experiments or discussions on it.

*We did not focus on the hydrologic units in this study because we used physics-based models, which provide a spatial distribution of hydrologic variables. We, however, study the hydrologic responses at the watershed scale. Indeed, for each of the hydrologic variables we simulated (snow water equivalent, infiltration, evapotranspiration, surface water, and groundwater), we compared the results at both watershed scale and single point. We clearly explain in the manuscript the differences at both scales.*

2. Moreover, the accuracy of the WRF meteorology model and the downscale and upscale techniques are very important to this paper＇s results. The authors validated the quality of different resolution forcing data at the watershed lumped level and the distributed level in Appendix A. However, in Figure A.2 and Figure A.3, the meteorology at 40.5km look bad as the pattern becomes very blurry. I would encourage trying a finer resolution, such as 27km. In addition, is it possible to validate the WRF and downscaled

meteorology with measured precipitation and temperature data for this research area? This will better valid your modelled and calculated meteorology forcing.

*We performed simulations with WRF data at 40.5 km resolutions because this is the resolution close to the global climate models (their resolution is ~50 km and above). The resolution is indeed very coarse but it is that resolution that most of the hydrologic models are using to project the evolution of hydrology in the future.*

*The comparison between WRF outputs and ground observations is not relevant to the conclusions of this study as these conclusions are in essence derived from comparisons between (equivalent) forcing data produced at different resolutions using nested-domain configuration of WRF. Validation of WRF could be relevant in the context of consistency of physics represented in the WRF in terms of atmospheric and land surface processes and their interactions. As mentioned in the manuscript, the WRF configuration used for this study has been tested against a variety of ground-based observational datasets could be found in previous publications by the authors. However, to address this concern of the author, in the revised version of the manuscript, we further elaborate these established validations.*

3. Title issue. It would be better to change "hydrologic metric" to something more ap- propriate. I suggest this because some hydrologists may understand "hydrologic met- ric" as hydrology model performance metrics, such as Nash–Sutcliffe efficiency (NSE) or Kling–Gupta efficiency. An alternative could be hydrologic prediction. Moreover, is the first word "on" redundant? Please correct me if I'm wrong.

*We agree with reviewer and we changed the title: "Sensitivity of meteorological forcing resolution on hydrologic variables"*

4. Presentation of the results. The writing of this paper needs to be improved. I found it hard to understand some sentences, the definitions of several terms and the full name of an abbreviation are missing, and some citations and captions are not standardized. Please see below for details.

*We have made a genuine effort to thoroughly check the manuscript for areas of writing improvement.*

5. In section 2a, I suggest specifying the areas of the Sierra Nevada Mountains and the Central Valley in Figure 1.

*Agreed. We now specify the two main areas of the watershed in the revised manuscript.*

Technical corrections

1. Line 268, "5 spatial resolutions". Change 5 to five. Similarly, to line 271.

*Changed.*

2. Equation 7 and Equation 8, are the two φ the same? I guess one is surface pressure head, and nother is subsurface pressure head. Please be consistent with the terms in Equation 1 and Equation 2, and be specific.

*Yes, both represent pressure-head. Parflow-CLM is an integrated hydrologic model as such the surface and subsurface pressure heads are the same. Please refer to lines 174 to 197 of the original manuscript for more details.*

3. Line 341, citation (Rasmussen et al., 2011) should be Rasmussen et al. (2011).

*Changed.*

4. Unclear sentences list. a. Lines 330-332 b. Lines 369-370 c. Lines 493-495 d. Lines 523-524 e. Lines 527-528 f. Lines 540-544 g. Lines 624-626 h. Lines 630-632

*We thank the reviewer for these specific points of potential improvement in the manuscript. We have modified each to make them clearer.*

5. Incorrect caption citation list. a. Line 347, Equation 5 -> Equation 6. b. Line 686, figure xx. c. Line 687, 0.01 C.

*We changed the caption.*

6. Undefined abbreviation: Line 342, WY.

*We now define WY (which means Water Year).*

7. Unclear referent. a. Line 561, the latter. b. Line 590, the latter. 8. Line 666, Climate Models -> Climate models.

*We changed these lines of the revised manuscript for clarity.*

---

## Author Response (AR1)

**Comments in plain text, response to comments in *blue italics*.**

**Editor**

Dear Authors,

Both referees of your paper suggest that major revisions are necessary, and therefore I invite you to revise your paper and resubmit after addressing the reviewer comments. The paper will be returned to the referees for a second review.

*We have thoroughly and carefully revised the manuscript to address the comments outlined by each of the reviewers and feel that it is now suitable for publication in HESS.*

I encourage you to seriously address the reviewer comments, which will help to improve the quality of the manuscript. For example, both reviewers ask for comparisons with observed data and such a comparison should be included in the revised manuscript.

*We have now added comparisons between measured and simulated precipitation and temperature at four selected stations; please refer to Appendix A3 in the revised manuscript. As indicated in the responses to the reviewers, we did not add these comparisons in the original manuscript because the main scope of this study is to assess how different spatial distributions of the same mass of precipitation, for example, impact hydrologic variables.*

Another instance that I noted in your response where you declined to answer was when one of the referees asks for advice on how your guidance on grid resolution should be applied for non-grid-based models: again, this is a reasonable question and should be answered.

*We have answered the reviewer question in the response to reviewer comments, and also added general guidance for the impact of forcing resolution on non-grid-based models in the conclusion section of the revised manuscript.*

*Our study is based on a grid-based integrated hydrologic model, which is a category of models widely used in studies of watershed dynamics because they allow for one to assess water and energy cycles from the subsurface to the lower atmosphere both in time and space. Non-grid based models are much more limited, namely due to their lack of physical representation of key system processes. While the results drawn from this study are mainly applicable to integrated hydrologic models, we, do analyze global values of integrated hydrologic model responses in this study, and show that in general the resolution of forcing does not significantly impact these signals at a watershed scale. Resolution of forcing mostly matters at point scales (i.e. when using grid-based models) not watershed to catchments scale due to the counterbalancing of under- and over- estimation of the forcing in space.*

I look forward to receiving your revised manuscript,
Best wishes
Hilary McMillan

**Anonymous Referee #1**

General comments

This paper tested the sensitivity of the spatial resolution of meteorological forcing data on hydrologic model results. The paper addresses a classic scientific question which is within the scope of HESS. The descriptions of experiments and calculations are complete. However, the idea and findings of this paper are not novel. Moreover, the conclusions are derived from one meteorology model and one hydrologic model, the results to me are insufficient to support the conclusions. Lastly, the paper is not well written and the presentation of results analysis is unclear.

*We thank the referee for acknowledging the scientific question highlighted in our work, as well as the experiments and the computations we performed. To the best of our knowledge, there aren't any existing studies, which utilize physics-based models to simulate both atmospheric and hydrologic processes in this coupled fashion with the goal to understand biases in resolution on hydrologic flow variables. This contribution is certainty novel.*

*We have thoroughly revised the manuscript to highlight the novelty of this approach and the importance of the study's findings. We have also thoroughly revised the manuscript for clarity and worked to make the writing more succinct and clear. We feel that these changes have improved the manuscript and made it more suitable for publication in HESS.*

Specific comments

1. Supportability of the findings. As mentioned above, the conclusions are derived from one meteorology model and one hydrologic model. These are not sufficient to provide a general conclusion. For example, the conclusion that ". . . the meteorological data should be at the resolution of the input data as well as the physics-based model to ensure a good precision and accuracy in the representativity of the snow dynamics." (Lines 333-335) may not be applicable if modelers adopt hydrologic response units, not grid cells, to build hydrologic models. For irregular computational unit based hy- drologic modeling, what is the appropriate input data resolution? I hope to see more experiments or discussions on it.

*We did not focus on hydrologic units in this study because we used physics-based models, which provide a spatial distribution of hydrologic variables. We do, however, study the hydrologic responses at the watershed scale via global variables. For each of the hydrologic variables we simulated (snow water equivalent, infiltration, evapotranspiration, surface water, and groundwater), we compared the results at both the watershed scale and at single points in space. We clearly explain in the manuscript the differences at both scales. In the conclusions section of the revised manuscript, we have now added that we found that the aggregated hydrologic response is less sensitive to the spatial resolution of forcing when compared to the grid-based models (please refer to lines 637-642 of the revised manuscript).*

2. Moreover, the accuracy of the WRF meteorology model and the downscale and upscale techniques are very important to this paper's results. The authors validated the quality of different resolution forcing data at the watershed lumped level and the distributed level in Appendix A. However, in Figure A.2 and Figure A.3, the meteorology at 40.5km look bad as the pattern becomes very blurry. I would encourage trying a finer resolution, such as 27km. In addition, is it possible to validate the WRF and downscaled

meteorology with measured precipitation and temperature data for this research area? This will better valid your modelled and calculated meteorology forcing.

*We performed simulations with WRF data at 40.5 km resolutions because this is the resolution close to the global climate models (their resolution is are typically 50 km or larger). The resolution is indeed very coarse, but it is that resolution that most of hydrologic models utilize to project the evolution of hydrologic processes in the future. This, in part, lies the motivation for this work.*

*The comparison between WRF outputs and ground observations is not relevant to the conclusions of this study as these conclusions are in essence derived from comparisons between (equivalent) forcing data produced at different resolutions using the nested-domain configuration of WRF. However, as mentioned in the manuscript, the WRF configuration used for this study has been tested against a variety of ground-based observational datasets, which is further explained in detail in previous publications by the authors (see citations listed in the revised manuscript on lines 241-243). Nevertheless, to address the reviewer's concern, we have now added comparisons between measured and simulated precipitation and temperature at four selected stations. Please refer to Appendix A3 in the revised manuscript.*

3. Title issue. It would be better to change "hydrologic metric" to something more ap- propriate. I suggest this because some hydrologists may understand "hydrologic met- ric" as hydrology model performance metrics, such as Nash–Sutcliffe efficiency (NSE) or Kling–Gupta efficiency. An alternative could be hydrologic prediction. Moreover, is the first word "on" redundant? Please correct me if I'm wrong.

*We agree with reviewer and we changed the title: "Sensitivity of meteorological forcing resolution on hydrologic variables"*

4. Presentation of the results. The writing of this paper needs to be improved. I found it hard to understand some sentences, the definitions of several terms and the full name of an abbreviation are missing, and some citations and captions are not standardized. Please see below for details.

*We have made a genuine effort to thoroughly check the manuscript for areas of writing improvement.*

5. In section 2a, I suggest specifying the areas of the Sierra Nevada Mountains and the Central Valley in Figure 1.

*Agreed. We now specify the two main areas of the watershed in the revised manuscript.*

Technical corrections

1. Line 268, "5 spatial resolutions". Change 5 to five. Similarly, to line 271.

*Changed.*

2. Equation 7 and Equation 8, are the two φ the same? I guess one is surface pressure head, and nother is subsurface pressure head. Please be consistent with the terms in Equation 1 and Equation 2, and be specific.

*Yes, both represent pressure-head. Parflow-CLM is an integrated hydrologic model as such the surface and subsurface pressure heads are the same. Please refer to lines 174 to 197 of the original manuscript for more details.*

3. Line 341, citation (Rasmussen et al., 2011) should be Rasmussen et al. (2011).

*Changed.*

4. Unclear sentences list. a. Lines 330-332 b. Lines 369-370 c. Lines 493-495 d. Lines 523-524 e. Lines 527-528 f. Lines 540-544 g. Lines 624-626 h. Lines 630-632

*We thank the reviewer for these specific points of potential improvement in the manuscript. We have modified each to make them clearer.*

5. Incorrect caption citation list. a. Line 347, Equation 5 -> Equation 6. b. Line 686, figure xx. c. Line 687, 0.01 C.

*We changed the caption.*

6. Undefined abbreviation: Line 342, WY.

*We now define WY (which means Water Year).*

7. Unclear referent. a. Line 561, the latter. b. Line 590, the latter. 8. Line 666, Climate Models -> Climate models.

*We changed these lines of the revised manuscript for clarity.*

**Anonymous Referee #2**

General comments:

The manuscript addresses the sensitivity of hydrologic variables to the spatial resolution of meteorological forcing inputs. Analysis of multiple components of the hydrological cycle in time and space makes the current study fit within the scope of HESS.

*We thank the reviewer for acknowledging the importance of this study and its fit within HESS.*

While the overall objective is to compare how the resolution of meteorological forcing data impacts hydrologic variables, it would be helpful to see how WRF model output from the simulations compares to actual observations.

*The comparison between WRF outputs and ground observations is not relevant to the conclusions of this study as these conclusions are in essence derived from comparisons between (equivalent) forcing data produced at different resolutions using the nested-domain configuration of WRF. However, as mentioned in the manuscript, the WRF configuration used for this study has been tested against a variety of ground-based observational datasets, which is further explained in detail in previous publications by the authors (see citations listed in the revised manuscript on lines 241-243). Nevertheless, to address the reviewer's concern, we have now added comparisons between measured and simulated precipitation and temperature at four selected stations. Please refer to Appendix A3 in the revised manuscript.*

The study year is said to be the wettest on record, so the WRF simulations are not being performed for a typical year, but rather one that lies in the tails of the distribution. As such, are there inherent errors associated with simulating an anomalous case versus typical?

*We acknowledge the reviewer's point, and have added further discussion of the topic in the revised manuscript. Namely, the choice of a wet year was intentional in this work, in order to understand how changes in meteorological forcing may present themselves in a "worst-case" scenario. The errors associated with inaccurately representing precipitation will be most obvious in the simulations presented here, and thus are considered a conservative estimate of the bias associated with overly coarsened meteorological forcing in hydrologic simulation.*

Comparison to observations may support whether WRF simulations are similar to reality, such that any biases are recognized before simply comparing model simulations to one another.

*Please see our response above. We have ensured that the model does not have any biases, given it's good performance when compared to observations.*

Some of the background is given without any supporting sources. One example is the paragraph from lines 107-130.

*We have added more references in the background section and in the paragraph pointed out by the reviewer to support our statements.*

While some of the information may seem like common knowledge, it is still important to state where this information came from. For example, in lines 114-115, the authors state something is ranked "among the highest in the world." This is highest according to what?

*Agreed. We added the reference in the revised manuscript.*

Similarly, lines 117-118 state that the majority of water resources in the region originate from snowmelt. How do we know this? Many readers will likely be familiar with such concepts, but those of interdisciplinary backgrounds may not be.

*We now add a reference to this statement. Please refer to the revised manuscript.*

The Paper is generally well organized, however Lines 82 and 83: Why does this study matter to the current work? It ties in later, I think, but it would be helpful to quickly tie your literature review into the current work.

*We thank the reviewer for acknowledging the good structure of this paper.*
*In lines 85 to 93, we link the literature review to the current scope of this work to highlight both the importance and the novelty of the current study.*

Lines 114-115: Please reword. I'm not sure what exactly is ranked highest in the world. Is it the agricultural sustainability or the necessity for understanding water resources? Please also provide a source to support this conclusion.

*California's agricultural productivity is ranked among the highest in the world. This statement is derived from statistics of the California Department of food and agriculture, which we now state in the revised manuscript.*

Lines 127-128: What is the period of record when stating "wettest on record." How do the authors know this?

*2017 is the wettest year on record since 1895. We have added references to the revised manuscript to support this statement.*

Lines 227-232: The authors state that 2017 is representative of a wide range of weather conditions, but also that it is a climatological anomaly. Why isn't a year that is more near-normal used to demonstrate an annual range of meteorological conditions? Swain et al. (2018) suggests a greater number of rapid transitions from dry to wet periods is likely in future climate scenarios. This may support the use of water year 2017 if the authors intentions were to capture a year that rapidly transitions from dry to wet but that is not clear in the current draft. This is the paper cited above: https://doi.org/10.1038/s41558-018-0140-y

*We selected the wettest year on California record for a variety of reasons. As stated above, high precipitation values are meant to highlight any disparities in utilizing overly coarse forcing, and thus the differences in hydrologic variables with various scales of forcing will conservatively show this bias. A second reason for simulating 2017 is because these extremes will likely characterize the future climate in California. As we highlight in the introduction of the manuscript, this year has many atmospheric rivers*

*and due to the intensity of the precipitation associated with these rivers, it is of interest to assess the accurate resolution required to model the impact of these conditions which are characterized as extreme now but may be considered "normal" in the future.*

*We have now cited Swain et al 2018 in the revised manuscript to support our choice. Please refer to the introduction in lines 124-129 of the revised manuscript.*

Line 349: The authors state that error increases as resolution of the meteorological forcing increases. Does this mean that finer resolution, i.e., lower grid spacing results in greater error, or higher grid spacing (coarser resolution) does? Readers that are unfamiliar with numerical weather prediction models may get confused, and minor elaboration would ensure that the main point the authors are trying to communicate is not misinterpreted. It looks like this is touched on in lines 367 to 369, but it would be easier to interpret if these statements were all grouped together.

*We modified the sentence for clarity. The correct sentence should be: "The coarsest spatial resolution (i.e. 40,5 km) of forcings shows the highest errors"*

Lines 617-648: This part of the manuscript is well organized, and I appreciate how the main points are separated into individual bullets points.

*We thank the reviewer for acknowledging the organization of the paper.*

Lines 630-632: The statement that the meteorological model should have the same resolution as the hydrologic model is made here, however, it does not appear that any simulations of WRF were performed with the exact same resolution as the hydrologic model. Line 197 says that the horizontal resolution of the hydrologic model is 200 meters while the finest horizontal resolution WRF simulation was 500 meters, so how is this conclusion supported by the current work when there were still differences in resolution? If there are past studies that support such a conclusion, please add relevant citations.

*Point taken. We changed the text to: "The results obtained with the different spatial distributions suggest that meteorological data with **a resolution close to the one of the hydrologic model** is needed to accurately reproduce the Snow Water Equivalent (SWE) distribution as well as the total volume of SWE."*

Lines 666-668: This statement is somewhat unclear. Is this referring to what climate models do in general, or is this referring to how they will be employed in future work? Please reword the statement for clarity.

*Climate models are used to simulate the current or past meteorological conditions (like the simulations we performed in this study) and also to predict the future climate conditions. In this sentence, we are referring to the use of climate models to project into the future. Our future studies will assess the impact of the spatial resolution of forcing on the simulated hydrology using future climate projections instead of the past meteorological conditions.*

---

## Author Response (AR2)

**Comments in plain text, response to comments in *blue italics*.**

**Editor**

Dear authors,

The edits to your paper have been re-reviewed, and the referee found that you satisfactorily addressed the major review comments. Thank you for your thorough response to these comments. Please address the remaining minor queries regarding your manuscript. Kind regards,
Hilary

*We have revised the manuscript to address the minor comments outlined by the referee and feel that our paper is now suitable for publication in HESS.*

**Anonymous Referee #1**

The revision has addressed most of my previous comments. I like the addition of the comparison with observed data (Appendix A3). I am also happy seeing that the authors looked at the hydrologic response at the watershed scale versus at the grid scale. This definitely enriched the findings of this paper.

*We thank the referee for their comments. We have addressed all their minor comments; see below.*

However, I'm not quite satisfied with the authors' response to the impacts of gridded forcing resolution on non-grid-based hydrologic models. I would like to point out that non-gridded hydrologic models can be physics-based and can provide distributed hydrologic variables. For example, the hydrologic response unit (HRU) based hydrologic models: the cold regions hydrological model (CRHM) (Pomeroy et al., 2007) and the structure for unifying multiple modeling alternatives (SUMMA) (Clark et al., 2015a, 2015b). Therefore, the grid size of hydrologic models can be taken in a more flexible way. It actually refers to the size of the basic hydrologic modeling unit. In this research, the authors take the grid-based hydrologic model as an example and conclude that the resolution of meteorological data is better to be close to the grid size of hydrologic models. This conclusion may be valid to non-grid-based hydrologic models, too, and could be a research avenue.

*We agree with the referee that the findings of this study might be limited to the grid-based integrated hydrologic models, which is also the main purpose of our work. Nonetheless, we have attempted to provide guidance by studying the aggregated hydrologic response at the watershed scale. However, as*

*suggested, future works could assess the impacts of the spatial distributions of forcing on HRU based hydrologic models and could compare the sensitivity of the two types of models to the spatial distributions of meteorological forcings. We have added a sentence regarding this future research venue in our conclusion, please refer to lines 671-673.*

Below are some specific comments.

1. Line 91. Change "…and meant..." to "are meant".
*Changed*

2. Line 110. Add "are" to the sentence "…, and periods of intense precipitation mainly caused by atmospheric rivers".
*We have removed the comma; the sentence does not need an "are"*

3. Figure 3 and Figure 5, y-axis tick labels need to be corrected. For example, -5.10^3 means -0.5*10^3, is it right?
*The y-axis labels are correct. As explained in the manuscript (section 4.2.), these values are large due to the small values of ET*

4. Line 257. Please specify Appendix A. I think it's "in Appendix A2".
*Changed*

5. Line 336. Please rephrase the sentence "thus small changes in ET are relatively large".
*Changed*

6. Line 379. Please specify Appendix A here. 1, 2, or 3?
*We refer to Appendix A2, we specified it in the manuscript.*

7. Line 442. Please rephrase the sentence "As shown in Figure 9 illustrating the spatial distributions of the absolute error of surface pressure-head, …".
*We have changed the sentence, now to read: "As shown in Figure 9 illustrating the spatial distributions of the absolute error of surface pressure-head (AE$\Psi$s), the percent error of the total surface water storage at the watershed scale is small because some regions in the domain over-estimate the pressure-head while others under-estimate the pressure-head"*

References:
Pomeroy, J. W., et al. "The cold regions hydrological model: a platform for basing process representation and model structure on physical evidence." Hydrological Processes: An International Journal 21.19 (2007): 2650-2667.
Clark, Martyn P., et al. "A unified approach for process‐based hydrologic modeling: 1. Modeling concept." Water Resources Research 51.4 (2015): 2498-2514.
Clark, Martyn P., et al. "A unified approach for process‐based hydrologic modeling: 2. Model implementation and case studies." Water Resources Research 51.4 (2015): 2515-2542.